# Pathoadapative Genomic Determinants of *Staphylococcus aureus* Community Skin Infections and Nasal Colonization

**DOI:** 10.3390/microorganisms13092023

**Published:** 2025-08-29

**Authors:** Cody A. Black, Wonhee So, Raymond Benavides, Julianne A. Mercer, Steven S. Dallas, James F. Shurko, Sarah M. Bandy, Benjamin A. Encino, Justina S. Lipscomb, Adriana Vargus, Christopher R. Frei, Grace C. Lee

**Affiliations:** 1College of Pharmacy, The University of Texas at Austin, Austin, TX 78712, USA; blackc1@uthscsa.edu (C.A.B.); mercerj3@uthscsa.edu (J.A.M.); encinob@uthscsa.edu (B.A.E.); lipscombj@uthscsa.edu (J.S.L.); freic@uthscsa.edu (C.R.F.); 2Joe R. and Teresa Lozano Long School of Medicine, The University of Texas Health at San Antonio, San Antonio, TX 78229, USA; dallass@uthscsa.edu; 3College of Pharmacy, Western University of Health Sciences, Pomona, CA 91766, USA; wso@westernu.edu; 4Department of Pathology and Laboratory Medicine, The University of Texas Health at San Antonio, San Antonio, TX 78229, USA; 5Department of Microbiology, University Health System, San Antonio, TX 78229, USA; 6South Texas Veterans Health Care System, San Antonio, TX 78229, USA

**Keywords:** *Staphylococcus aureus*, methicillin resistance, pathogenicity, genome-wide association study, skin and soft tissue infection, metabolic, community, colonization, purulent, bacterial genomics

## Abstract

*Staphylococcus aureus* is a leading cause of skin and soft tissue infections (SSTIs), yet the bacterial genomic adaptations underlying the transition from nasal colonization to invasive infection remain incompletely defined. We sequenced and analyzed 157 *S. aureus* isolates (126 from SSTIs and 31 from asymptomatic nasal colonization) from a primary care network in South Texas. Using genome-wide association studies, non-synonymous single-nucleotide variant (NSNV) profiling, and machine learning, we identified strain-specific adaptations in metabolic and regulatory pathways. SSTI isolates exhibited significant enrichment of nitrogen assimilation, purine biosynthesis, menaquinone production, and anaerobic respiration genes. Elevated copy number and colocalization of phage-linked metabolic genes—including *nirB*, *narH*, and *nifR3*—suggest a pathoadaptive genomic island supporting infection-specific energy generation. The enrichment of α/β-hydrolase domain-encoding genes was associated with clinical severity. To quantify severity, we developed the Purulent Ulcer Skin (PUS) score, which integrates wound size, drainage, and erythema. The α/β-hydrolase and lipoprotein genes were significantly associated with higher PUS scores (higher SSTI severity) and phage-encoded virulence gene products were linked to larger wound size. Machine learning prioritized *purL* and other metabolic loci as key infection classifiers. NSNVs and unitig-level changes co-localized within nutrient transport, stress resistance, and cytolytic genes, supporting a model of multi-layered genomic selection. Metagenomic assemblies of nasal microbiota were enriched for *Staphylococcus*, *Enterococcus*, and *Micrococcus* species, core metabolic pathways, and taxon-specific virulence determinants. This underscores the roles of metabolic and virulent co-networks within nasal commensals and their adaptive capacity for pathogenic transition. These findings provide a potential genomic blueprint of *S. aureus* pathoadaptation during SSTI and are a step towards the development of novel therapeutic targets.

## 1. Introduction

*Staphylococcus aureus* is a versatile organism that toggles between a benign colonizer and a dangerous pathogen [1]. While one-third of healthy individuals carry *S. aureus* asymptomatically in their nasal passages, the bacterium remains a leading cause of skin and soft tissue infections (SSTIs), sepsis, and invasive diseases globally. Understanding what drives this shift from colonization to infection is central to developing effective therapies and diagnostics as well as prevention efforts.

The debate over whether this transition is purely host-driven or also driven by pathogen-intrinsic traits remains unresolved. While some evidence suggests that all colonizing strains can cause disease under permissive conditions, other studies point to strain-specific virulence factors [2,3]. Approximately 70–90% of *S. aureus* SSTIs are caused by strains genetically matching the patient’s own colonizing isolates [4]. The high genetic similarity between colonizing and infecting isolates suggests a combined role for host and microbial factors.

Metabolic adaptations for survival in nutrient-limited infection environments are particularly important, with nitrogen assimilation and anaerobic respiration pathways showing strong associations with degree of virulence [5]. Similarly, the role of prophage-encoded virulence factors has gained increasing recognition, as has the importance of lineage-specific genomic architectures in pathogenic potential [6]. However, these studies have typically examined either colonization or infection isolates exclusively, leaving critical gaps in our understanding of the direct genomic changes accompanying pathogenic transition.

Herein, we conducted a comparative genomic analysis of clinical *S. aureus* isolates from matched SSTIs and asymptomatic nasal colonization cases from primary care clinics in South Texas [7,8,9,10,11,12,13]. We used whole-genome sequencing, metagenomic and machine learning approaches to identify genetic and metabolic adaptations linked to invasive potential of *S. aureus*. Our findings highlight distinct but related genomic features—such as nitrogen assimilation clusters, purine biosynthesis genes, and prophage-encoded factors—that distinguish pathogenic from colonizing strains. These adaptations may serve as new targets for diagnostics and/or therapeutics aimed at preventing disease without disrupting the commensal flora.

## 2. Materials and Methods

### 2.1. Study Setting and Population

This study was conducted using clinical and bacterial isolate information from a cohort of patients with *S. aureus* SSTIs or nasal colonization within the South Texas primary care setting from 2007 to 2016 [7,8,9,10,11,12,13]. Patients were eligible for study enrollment if they provided informed consent, were 18 years of age or older, and presented to one of the participating clinics with an SSTI. Healthcare providers collected wound samples and patient information from each patient (e.g., demographics, infection characteristics, clinical information). As controls, *S. aureus* nasal colonization isolates were collected from patients who presented to these clinics without SSTIs, from February to May 2015. This study was reviewed and approved by the Institutional Review Board of the University of Texas Health Science Center at San Antonio and University Health System, San Antonio.

### 2.2. Bacterial Isolates and Microbiological Analysis

Previously, we identified the South Texas region to have a disproportionately higher incidence of SSTI and community-associated MRSA, offering a relevant population for strain-specific genomic analysis [7,8,9,10,11,12,13]. We selected 126 *S. aureus* isolates from SSTIs and 31 nasal colonization isolates for whole genome sequencing (WGS). Samples were plated on blood agar, incubated, and then sub-cultured on methicillin-resistant *Staphylococcus aureus* (MRSA)-selective agar (MRSASelect chromogenic agar plates; Bio-Rad Laboratories, Hercules, CA, USA) for phenotype confirmation. Identification and isolation of MRSA were confirmed using latex agglutination and cefoxitin phenotypic tests. Susceptibility testing of study isolates to fourteen antimicrobials was conducted with Vitek 2 AST-GP75 cards, following CLSI guidelines M100-S15 (2015).

### 2.3. DNA Sequencing and Analysis

For WGS, total bacterial DNA was extracted using a DNeasy PowerSoil kit (Qiagen, Redwood City, CA, USA). WGS was conducted on all isolates using a NextSeq 500 sequencing instrument (Illumina Inc., San Diego, CA, USA) with 150-base paired-end reads (UT Health San Antonio, San Antonio, TX, USA), as previously described [11].

### 2.4. Contig Assembly, Unitig Generation, de Bruijn Graph Construction and Variant Calling

Raw paired-end FASTQ files were first trimmed using fastp (v0.23.0) to remove low-quality bases and adapters, followed by quality assessment using FastQC (v0.12.1). High-quality reads were assembled into contigs in FASTA format using SPAdes (v4.0.0), with the—careful mode enabled to reduce misassemblies and post-assembly polishing applied to enhance contig accuracy. All genomes passed key quality thresholds as evaluated by FastQC (Appendix A). All MLST, *spa* and key result genes were successfully assembled across the respective samples (Appendix A).

To identify non-synonymous nucleotide variants (NSNVs) from *S. aureus* strains from both infectious and nasal colonization sources, assembled genomes were aligned against two references using Snippy (v4.6.0): (1) NCTC 8325 (accession: CP000253.1), an ST8 infection-derived reference; and (2) ATCC 27,217 (accession: SRR5364258), an ST5 strain isolated from the nares of a nurse. Variant call files (VCFs) were annotated and parsed to extract genomic position, locus tag, and predicted variant effects. Only non-synonymous variants (missense, insertion, deletion, frameshift, premature stop) were retained for downstream analysis. Core genome alignments were generated using Snippy (v4.6.0), producing a concatenated alignment of variant positions without gaps for phylogenetic reconstruction via FastTree, which served as a similarity matrix to account for lineage structure in subsequent GWAS analyses. NSNV effects were converted to binary features for all genes across the entire genomes of the *S. aureus* collection (*n* = 157). These data were organized by source (SSTI vs. nasal colonization) and MLST order then visualized as a heatmap (Appendix A).

Unitigs were generated using unitig-counter (v1.1.0) (https://anaconda.org/bioconda/unitig-counter [accessed on 10 June 2024]), which builds on the DBGWAS framework. All *S. aureus* genome assemblies were provided as input via a strain list. A compressed de Bruijn graph was constructed using the GATB library, from which unitigs were extracted. The resulting unitig presence/absence matrix (unitigs.unique_rows.Rtab) was formatted for input into Pyseer (v1.3.12).

For the bGWAS, unitigs were tested for association with infection phenotype (SSTI vs. asymptomatic nasal colonization) using Pyseer’s linear mixed model (-lmm). A core genome phylogenetic distance matrix was incorporated to adjust for shared ancestry and control for clonal population structure. This lineage-aware model improves the detection of genotype–phenotype associations by accounting for population stratification.

Unitig distributions were visualized using boxplots, and genome-wide association statistics were calculated. A genome-wide significance threshold was determined using the number of unique unitig patterns over the total number of unitigs, resulting in 106,840 unitig patterns with a threshold of *p* < 4.68 × 10^−7^. This was applied to correct multiple testing and population structure. Genes mapped to significant unitigs and NSNVs were functionally annotated using DAVID for pathway and enrichment analyses.

### 2.5. Phage and Antimicrobial Resistance Gene Annotation

Prophage regions were first identified using the PHASTEST web server (https://phastest.ca/ [accessed on: 7 August 2024]), which detects and classifies prophage elements within bacterial genome assemblies based on gene content and sequence similarity. Assembled contigs in FASTA format were submitted, and predicted prophages were categorized as intact, questionable, or incomplete, with coordinates and associated phage information extracted. FASTA (fna) sequences corresponding to PHASTEST-identified prophage regions were then used for functional annotation with Pharokka (v1.2.1), a streamlined pipeline for phage genome analysis. To further investigate structural homology, we annotated these same phage regions using Phold (v0.2.0), which predicts structural domains and phage proteins based on homology to known viral proteins. In parallel, antimicrobial resistance (AMR) genes were annotated using staramr (https://github.com/phac-nml/staramr [accessed on 3 July 2025] v0.7.2), which screens genome assemblies against the ResFinder and PointFinder databases using default thresholds (≥90% identity, ≥60% coverage). Detected AMR genes, associated resistance phenotypes, and chromosomal point mutations were compiled to characterize the resistome of each sample.

### 2.6. Functional Analysis of Enriched Genes

Functional relevance was assessed through Gene Ontology Biological Process (GOBP) terms, allowing us to associate biological pathways with observed genetic variations. Genes associated with unitigs and NSNVs were mapped to Gene Ontology (GO) and Kyoto Encyclopedia of Genes and Genomes (KEGG) pathways using Database for Annotation, Visualization, and Integrated Discovery (DAVID) [14]. Pathways of interest, including nitrogen metabolism, purine/pyrimidine biosynthesis, and anaerobic pathways, were assessed for functional enrichment.

### 2.7. Lineage-Associated Unitig and Restriction Modification (RM) Type Distribution

RM types I-IV were analyzed by aligning whole-genome sequences against REBASE (https://rebase.neb.com/rebase/rebase.html [accessed on 3 July 2025]) reference sequences using BLAST v2.12.0, with only top hits (≥80% alignment) retained. Presence-absence matrices were created to examine RM gene distribution by clonal complex (CC).

### 2.8. Whole Genome Copy Number Variation (CNV) Analysis

Processed reads were aligned to the *S. aureus* NCTC8325 reference genome (GenBank accession: CP000253.1) using BWA-MEM (v0.7.18) with default settings. Resulting SAM files were converted to sorted and indexed BAM files using SAMtools (v1.9). Per-base read depths were computed using BEDTools genomecov with the -dz option. A custom BED file containing gene coordinates and locus tag annotations was used to map coverage to specific loci. For each gene, average read depth was calculated and normalized to the average depth of seven Multilocus Sequence Typing (MLST) housekeeping genes: *arcC*, *aroE*, *glpF*, *gmk*, *pta*, *tpi*, and *yqiL*. CNV for each locus was calculated as: CNV = (mean locus depth)/(mean depth of MLST genes). Comparing CNVs between samples (e.g., A1 vs. A1016), average CNV values per locus were computed and differences were assessed by both raw and absolute CNV differences.

### 2.9. Machine Learning Modeling

XGBoost was implemented using the XGBClassifier from the XGBoost library (version 3.0.0). The dataset was split into training (70%) and testing (30%) sets using the train_test_split function from the sklearn.model_selection module, with a random seed of 42 to ensure reproducibility. The model was trained on the training set using the default hyperparameters, with eval_metric = ‘logloss’ specified for binary classification. After training, the model’s performance was evaluated on the test set using accuracy and the area under the receiver operating characteristic curve (ROC AUC) as primary metrics. Feature importance was initially determined using the F-score, which reflects the number of times a feature was used to split data across all decision trees in the model. To further interpret the model, SHAP (SHapley Additive exPlanations) values were calculated using the SHAP library. SHAP values provide a more detailed understanding of how each feature contributes to individual predictions. Missing lesion sizes (10/126) were imputed with an XGBOOST machine learning model using caret (v6.0-94) with transformed binary data for 111,490 unitigs significantly associated with infection or colonization across all 157 isolates. The colonization isolates (*n* = 31) were given a wound size of zero centimeters.

### 2.10. Post-GWAS Unitig Stratified by Clinical Severity

To refine candidate infection-associated variants based on clinical severity, we developed a composite purulent cellulitis severity score—the ‘Purulent Ulcer Skin’ (PUS) score. The score is derived by applying a gradient system: wound size (centimeters) measurements receive 1 point if between 0 and 4.9, with an additional point added for every additional centimeter increase; temperature follows a similar gradient, with points increasing as temperature exceeds 101.2 °F. Drainage and erythema are each assigned 1 point if present, with a maximum of 1 point awarded even if both are present. Also, candidate unitigs with mapping quality (mapq) scores > 30 were prioritized to ensure uniqueness and alignment confidence. Stratified unitigs were mapped back to the genome from which it was derived (de novo assembled contigs) to identify associated genes or genomic regions. Functional annotations from Bakta were evaluated, and literature searches were conducted to investigate the potential role of top candidates in *S. aureus* biology.

### 2.11. Metagenomic Binning and Annotation of Nasal Swab Microbiota

Seventeen nasal swab samples from colonized individuals underwent metagenomic processing. Raw reads were first trimmed using Fastp and quality assessed via FastQC. All 17 trimmed FASTQ files were co-assembled into scaffolds using metaSPAdes (v3.15.3) with default parameters. Reads from each sample were mapped back to the assembled scaffolds using Bowtie2 (v2.4.5) to generate BAM files. A depth matrix was created from the BAM files, which served as input for MetaBAT2 (v2.15) for contig binning. MetaBAT2 was run using default settings, including a minimum contig length of 1500 bp.

CheckM (v1.2.2) was used to assess bin quality, revealing that bins 2 through 9 met high-quality standards—defined as ≥90% completeness and ≤5% contamination. Bin 1 was excluded due to low quality. Annotation of these bins was performed using Bakta (v1.6.1) in both standard and meta modes to enhance gene prediction in metagenomic data.

### 2.12. Statistical Analysis

Descriptive statistics were used to classify patient characteristics and the genetic profile of the bacterial isolates. The chi-square test or Fisher’s Exact test was used to compare categorical variables between SSTI and colonization groups. Student’s *t* test or Wilcoxon Rank Sum test were used for continuous variables. An alpha level of 0.05 was used to detect statistical significance. Statistical analyses were performed using R v4.2.2 and Python v 3.12.10. RM-type distribution was analyzed using a Mann–Whitney U test and visualized as relative contributions by CC.

A z-score transformation was applied to the effect sizes to normalize and compare across unitigs and NSNVs. For each unitig or NSNV, the z-score was calculated based on the deviation from the mean effect size, standardized by the standard deviation. This allowed us to rank unitigs and NSNVs based on their deviation from the population mean, thereby highlighting variants with particularly strong associations. Variants with high positive z-scores indicate strong effects above the mean, while those with high negative z-scores represent strong effects below the mean. Variants with z-scores > 1.96 or ≤1.96 were highlighted as potentially significant, warranting further investigation.

For CNV analysis, statistical significance was determined using paired *t*-tests, with false discovery rate (FDR) correction applied for multiple comparisons. Loci with adjusted *p*-values < 0.05 were considered significant. Fold change was calculated as the ratio of CNV values between samples for each locus.

### 2.13. Data Availability

The raw whole-genome sequencing data have been deposited in the NCBI Sequence Read Archive (SRA) under BioProject accession number PRJNA1246601. Outside of the Pyseer environment, all code used in the genome-wide association and machine learning analyses are available at https://github.com/codyadamblack/starnet-infection-gwas [accessed on 3 July 2025].

## 3. Results

### 3.1. Study Population and Clinical Characteristics

We sequenced 157 *S. aureus* isolates recovered from 157 patients (126 from SSTIs and 31 from nasal colonization) presenting to participating primary care clinics in South Texas [7,8,9,10,11,12,13]. The clinical characteristics of the individuals with *S. aureus* SSTIs and nasal colonization are shown in Appendix A. Overall, the cohort had an average age of 41 years (SD ± 13), half were male, and majority reported to be Hispanic/Latino (76%). A significantly higher proportion of nasal carriers reported being healthcare providers (*p* < 0.01). All other characteristics were similar across the two groups.

### 3.2. Distinct Clonal Complex and MRSA Distributions Between SSTI and Colonization Isolates

We found significant differences (*p* < 0.01) in the distribution of clonal complexes between isolates from nasal carriers and those isolated from SSTIs (Figure 1A, Appendix A). A significantly higher proportion of the SSTI strains were of CC8 compared to carrier strains (68% vs. 10%; *p* < 0.01). Comparatively, a significantly higher proportion of carrier strains were CC30 (23% vs. 0%), CC45 (16% vs. 2%), and CC5 (23% vs. 2%) compared to SSTI strains (*p* < 0.01 for all comparisons). Furthermore, MRSA was significantly more common in SSTI samples (63%) compared to nasal carriers (13%) (*p* < 0.01).

The distribution of *S. aureus* isolates in Appendix A shows a predominance of CC8 (88 isolates), with other clades, such as CC5, CC45, and CC30, represented in smaller numbers. Plasmid profiles highlight the prevalence of *rep16* (123 isolates), *rep7c* (115), and *rep19* (98), suggesting these plasmid types are common across isolates. Key antimicrobial resistance genes include *blaZ* and *mecA* (85), with other genes like *mph (C)* and *msr (A)* also frequently present. Staphylococcal Cassette Chromosome mec (SCC*mec*) types indicate a predominance of *SCCmec Type IVa* (81 isolates). Regarding *spa* types, t008 was the most common (36 isolates), followed by t622 (21) and several others, illustrating a diverse *spa* profile within the collection.

Within selected clonal complexes, unitig diversity also varied (Figure 1B). CC8 strains demonstrated the widest range, while CC30 and CC6370 had lower median counts. This variation suggests that certain clonal complexes might harbor a greater diversity of unitigs, reflecting lineage-specific genomic adaptations. These findings indicate that both infection phenotype and lineage may contribute to the genomic diversity within *S. aureus* populations.

### 3.3. Clonal Complex-Dependent Restriction-Modification System Signatures Suggest Ecological Specialization

Analyses of RM system distribution across *Staphylococcus aureus* CC (Figure 1D) revealed distinct patterns in the prevalence of RM types (I-IV). RM-I systems were dominant in most CCs, particularly in CC25, CC12, CC20, and CC1159, where RM-I contributed between 87.8% and 97.3%, underscoring its key role. However, variation emerged with RM-II and RM-IV across different MLSTs. For instance, ST59 and CC8 showed notable RM-II contributions (50.9% in CC8), while RM-IV was particularly prominent in CC4236 (63.3%) as well as present in CC59 (37.7%) and CC30 (38.2%). Notably, RM-III was nearly absent across clonal complexes, except for minor presence in CC20 and CC45, suggesting its limited functional role within these strains. These findings suggest that CC-specific reliance on RM types may reflect unique evolutionary pressures or functional adaptations, with strains like CC4236 exhibiting a dependence on RM-IV, contrasting with the RM-I dominance observed broadly across other sequence types.

### 3.4. CNV Profiling Identifies Elevated nifR3 and Phage-Linked Genes in SSTI Isolates

To identify genomic loci associated SSTI versus nasal colonization, we performed whole-genome CNV analysis using normalized read depth data across 2969 *S. aureus* genes annotated against the NCTC 8325 reference genome (Appendix A). CNVs were computed by normalizing gene-level read depths to the mean coverage of seven MLST housekeeping genes per sample. Comparative analyses were conducted between isolates from SSTI (*n* = 126) and nasal colonization (*n* = 31) samples.

A total of 1048 genes demonstrated significant copy number differences between the two phenotypes after FDR correction (FDR < 0.05). Cohen’s *d* effect sizes ranged from –1.96 to +1.63, where positive values indicate higher CNVs in SSTI isolates and negative values suggest higher CNVs in colonization isolates. Among the top genes with the highest positive effect sizes (Cohen’s *d* ≥ 0.75, Figure 2), several encode phage-related or virulence-associated functions. For example, SAOUHSC_00744, encoding the nitrogen regulation protein NIFR3, had the strongest positive effect (Cohen’s *d* = 1.63, FDR = 1.3 × 10^−4^), with significantly elevated CNVs in SSTI isolates. Other highly enriched genes included a site-specific DNA-methyltransferase (SAOUHSC_00397, *d* = 1.62, FDR = 1.6 × 10^−6^), a staphylococcal superantigen-like domain-containing protein (SAOUHSC_00391, *d* = 1.36), and fibronectin binding protein B (SAOUHSC_02802, *d* = 1.16). Notably, these genes include elements involved across host immune modulation, bacterial adhesion, and phage interactions.

Conversely, genes significantly enriched in colonization isolates (Cohen’s *d* ≤ −0.75) were associated with stress response, transcriptional regulation, and membrane transport. For instance, SAOUHSC_02814, encoding an integral membrane protein, and SAOUHSC_01149, a cell division protein FtsA, showed lower CNVs in SSTI isolates (Cohen’s *d* = −0.75 and −0.75, respectively). One of the most negatively associated genes was SAOUHSC_01704, a bacterial toxin domain-containing protein, with a Cohen’s *d* of −1.96, highlighting its strong relative depletion in SSTI. Across the genome, there was a trend towards a greater number of genes with increased CNV in colonizing isolates compared to those from SSTI (Appendix A).

Together, these data suggest that both mobile genetic elements and specific core genome loci exhibit altered copy number dynamics that differentiate SSTI-causing *S. aureus* strains from colonizing strains. Genes associated with phage biology, adhesion, and immune evasion were particularly enriched in infection isolates, while regulators and membrane-associated proteins appeared more prevalent in colonizers. These findings point to a potential genomic CNV signature underlying pathoadaptation during purulent *S. aureus* infection.

### 3.5. Multi-Layered Genomic Association Studies Reveal Pathoadaptive Signatures

#### 3.5.1. Unitig-Based bGWAS Highlights Metabolic and Virulence Loci Associated with SSTI

The heritability estimate for the SSTI vs. colonization phenotype was 0.33, consistent with a moderate-low heritability of the phenotype. All 157 genomes yielded 382,883 unitigs, which were loaded into Pyseer. Of these, 79,463 were pre-filtered due to null patterns, e.g., presence across all samples, leaving 303,420 unitigs which were tested, printed, and visualized (Appendix A). The produced Q-Q plot (Appendix A) lacked *p*-value shelving (i.e., uniform distribution of *p*-values near the null) indicating minimal inflation due to population structure, which is also controlled in Pyseer via phylogenetic distance matrices. 303,420 unitigs were annotated using a library of thirty-two reference *Staphylococcus aureus* strains (Appendix A). Most annotations were from the reference genome NCTC_8325 as this strain was the first within the unitig annotation workflow. This resulted in 299,255 total annotated unitigs, 235,529 of which were located within a gene of a reference strain. Of the remaining 63,726 unitigs, 17,911 were salvaged by annotating with Bakta (v5.1), resulting in a total annotated unitig library of 253,411 which mapped to 8977 annotated regions.

The unitig count distribution varied significantly between the SSTI and colonization groups (Figure 1C). Overall, 57,767 unitigs were unique to SSTI and 24,126 were unique to colonization, while 300,990 unitigs were shared between both groups; among these, 160,657 were present in at least 20% of isolates in each phenotype. The SSTI isolates exhibited a wide range of unitig counts, with some samples showing notably high counts (e.g., sample SAMN47787180 with 146,175 unitigs), while most colonization samples had significantly lower counts. Specifically, SSTI samples contained an average of 168,974 distinct unitigs per sample compared to 156,842 in colonization samples (average difference of 12,132 unitigs). The higher average unitig count in SSTI samples suggests a potential link between elevated unitig counts and the infection phenotype. This may imply that samples associated with infection tend to have more diverse or complex genomes, due to greater genetic variability linked to virulence or adaptability. Notably, one SSTI isolate (SAMN47787133; CC8-like) and four colonization isolates (SAMN47787249/CC5, SAMN47787255/CC8, SAMN47787237/CC8, and SAMN47787245/CC8) showed significant enrichment in phenotype-shared unitigs (165,493–172,295). This disproportionate accumulation of shared genomic features may reflect a genetic predisposition to infection in these colonization isolates.

#### 3.5.2. Functional Enrichment Links Purine, Thiamine, and Menaquinone Pathways to Invasive Phenotype

To identify functional categories enriched in *S. aureus* genes associated with SSTI, unitig-based association statistics were merged with KEGG pathway hierarchies. A total of 59 unitig-mapped genes met the filtering thresholds of a lineage-adjusted −log_10_ *p*-value ≥ 6.3 and an average effect size (β) ≥ 0.5 (Appendix A). These genes were mapped to multiple KEGG biological categories, including metabolism, environmental information processing, and cellular processes (Figure 3).

Within metabolic pathways, several genes involved in purine biosynthesis exhibited strong statistical associations with SSTI. These included *purL* (Q2FZJ0), *purK* (Q2FZJ4), *purF* (Q2FZJ3), and *purM* (Q2FZJ2), which displayed average effect sizes ranging from 0.56 to 0.60, average allele frequencies (AF) > 0.60, and −log_10_ *p*-values above 14. *purL* showed the highest allelic diversity (AF–MAF [minor allele frequency] = 0.25), indicating strong directional selection for its major allele in infection-associated strains. Although we controlled for lineage effects using core genome distances, residual confounding by clonal background cannot be excluded. These purine-related genes are integral to supporting bacterial replication and nucleotide salvage during host colonization and infection as well as overlap with genomic regions within-host selection and metabolic fitness.

Genes participating in thiamine metabolism and amino acid biosynthesis were also highly enriched. *thiM* (Q2FWG2), involved in thiamine (vitamin B1) metabolism, had an average β of 0.54 and maxp (maximum −log_10_ lineage adjusted *p*-value) of 16.49, with an AF of 0.60 and MAF of 0.40. *thrB* (Q2FYV2), involved in threonine biosynthesis, showed a β of 0.56 and maxp of 15.96, further supporting the potential importance of vitamin and amino acid pathways in SSTI pathoadaptation.

Genes related to lipid and cofactor metabolism were also evident among significant loci. *menB* (Q2FZL5), encoding a key enzyme in menaquinone biosynthesis (vitamin K2), showed an average β of 0.54, AF of 0.57, and maxp of 11.38. *folE2* (Q2FXQ9), involved in tetrahydrofolate biosynthesis, was also above threshold with a β of 0.51 and maxp of 10.19. These genes may support anaerobic respiration and electron transport chain function, especially under oxygen-limited conditions in abscesses and deep tissue infections and correspond with findings from the anaerobic survival and carbon metabolism analyses described in earlier sections.

Bacterial toxin-associated genes were represented by genes related to both signal transduction and membrane transport. *hlgC* (Q2FVK2), a component of the γ-hemolysin pore-forming toxin complex, had an AF of 0.52, β of 0.53, and maxp of 17.07. *ftsY* (Q2FZ48), involved in protein translocation via the signal recognition particle pathway, exhibited β of 0.54 and maxp of 13.92. Additional transport-related genes such as *deoC* (Q2G224), a deoxyribose-phosphate aldolase, and *ptsL* (Q2G2R5), a component of the lactose-specific phosphotransferase system (PTS), were also retained with average β values > 0.52 and significant *p*-values. These pathways collectively enable sensing and adaptation to host-associated metabolic cues and potentially reflect coordinated regulation with other anaerobic, or carbon utilization pathways described in this study.

Among the genes associated with cellular regulation and virulence, *AgrD* (Q2FWM6), encoding a quorum-sensing peptide involved in virulence gene expression, stood out with a β of 0.55 and maxp of 13.68. *sspA* and *sspB*, coding for secreted proteases, also surpassed the effect size and *p*-value thresholds, further emphasizing the role of virulence modulation in SSTI strains. These genes complement regulatory and pathogenesis-related findings from both the lipase activity and protease variant analyses described in the discussion and supplemental sections.

Loci involved in nitrogen-related metabolism and amino acid transport were also identified. *gcvPB* (Q2FYU6), a component of the glycine cleavage system, and *argG* (Q2FWU7), involved in arginine biosynthesis, exhibited average β values above 0.52 with high AF and MAF diversity. These genes may support nitrogen scavenging and amino acid homeostasis in host tissues and are also referenced in the broader context of within-host adaptation and colonization potential.

All genes plotted in Figure 3 passed the thresholds for transformed *p*-value (≥6.3), average effect size (≥0.5), and average AF > 5%. Dot sizes in the figure reflect allelic diversity (AF−MAF), with genes like *purL*, *purK*, *hlgC*, and *ftsY* exhibiting high diversity values. Top effect size genes within each of the KEGG pathways include Q2FWG2 (*thiM*), Q2FZJ4 (*purK*), Q2FYV2 (*thrB*), Q2FVK2 (*hlgC*), Q2FZ48 (*ftsY*), Q2FWM6 (*AgrD*), Q2FYS9 (aconitate hydratase), Q2G2R5 (PTS lactose-specific EIIA), Q2G224 (*deoC*), and Q2FVL3 (ABC transporter permease). All fifty-nine loci are annotated in Appendix A and were further assessed for pathway-level enrichment using DAVID functional annotation of mapped KEGG terms.

#### 3.5.3. NSNVs Mapped to DNA Repair, Cell Wall, and Stress Response Genes

To determine whether the unitigs enriched in these biological processes are neutral or result in functional consequences within reference coding sequences, we evaluated NSNV across 2876 genes, using both an infection-sourced reference (NCTC 8325) and nasal colonization reference (ATCC 27217). Colonizing vs. SSTI sourced isolates were observed to have proportionately more genes with NSNVs (616 vs. 193 genes, respectively) when compared to NCTC 8325; additionally, NSNVs were found in 102 and 120 genes, respectively, when compared to ATCC 27217.

A heatmap of NSNVs across 157 *S. aureus* isolates, compared to the reference strain NCTC 8325, revealed a striking cluster of NSNV-enriched genes in SSTI-associated samples (Appendix A). These variants were concentrated in a distinct region of the heatmap, highlighted by red gene clusters, and were absent in colonization isolates. The affected genes were associated with bacteriophages, including those annotated as “phage”, “bacteriophage”, “autolysin”, “lysostaphin”, “PVL orf”, “integrase”, and “protease”. This pattern suggests that specific prophage-related genomic regions may contribute to the pathogenesis of SSTI.

Treemap analysis of enriched Gene Ontology (GO) terms revealed that NSNVs in *S. aureus* infection and colonization isolates were significantly associated with key biological processes, including proteolysis, cell wall organization, metal ion transport, and stress responses (Appendix A). The most highly enriched term was proteolysis (GO:0006508), indicating strong selective pressure on protein degradation pathways, which are critical for bacterial virulence, nutrient acquisition, and regulatory turnover. This cluster is further subdivided into related processes such as DNA repair (e.g., homologous recombination, mismatch repair), amino acid biosynthesis (e.g., arginine, leucine, tryptophan pathways), and tRNA modification, suggesting that NSNVs in these genes may influence bacterial adaptability in different host environments. Cell wall organization (GO:0071555) was another major functional cluster, underscoring the importance of structural integrity in *S. aureus* survival. Variants in genes involved in peptidoglycan synthesis and cell division (e.g., GO:0051301) were prominent, consistent with their roles in antibiotic resistance and host immune evasion. Additionally, nickel cation transport (GO:0015675) and related ion homeostasis pathways were enriched, potentially contributing to phenotypic divergence to host-imposed metal restriction during infection. Stress response mechanisms were also prominent, with significant enrichment in antibiotic response (GO:0046677), oxidative stress resistance (GO:0034599), and heat shock (GO:0009408). These findings suggest that the identified NSNVs support stress tolerance during host–pathogen interaction. Furthermore, siderophore biosynthesis (GO:0019290) was enriched which plays a role of iron acquisition in *S. aureus* pathogenicity.

KEGG pathway analysis (Appendix A) reinforced the central role of adaptive metabolic remodeling in infection-associated *S. aureus* isolates. Among the seventy-seven enriched pathways, amino acid metabolism was most prominent, comprising 23.3% of all terms, including biosynthesis of branched-chain (valine, leucine, isoleucine), aromatic (phenylalanine, tyrosine, tryptophan), and sulfur-containing amino acids. Carbohydrate metabolism represented 16.9% of pathways, highlighting glycolysis/gluconeogenesis, pyruvate metabolism, and the TCA cycle—hallmarks of carbon flexibility in nutrient-restricted environments. Though only ~2.6% of terms fell under lipid metabolism, the presence of glycerophospholipid and glycerolipid metabolism suggests remodeling of membrane components during host colonization and infection. Additional enrichment in cofactor biosynthesis (11.7%)—including pantothenate, CoA, and lipoic acid—points to enhanced metabolic self-sufficiency in SSTI strains. Broad terms such as “Metabolic pathways” (77 hits) and “Biosynthesis of secondary metabolites” (42 hits) underscore the global rewiring of central and specialized metabolism. These findings support the concept that *S. aureus* strains associated with SSTI exhibit a distinct metabolic repertoire that enables persistence and proliferation across diverse tissue microenvironments.

We explored genomic adaptations associated exclusively within the predominant lineage, CC8, resulting in 85 SSTI versus three nasal colonization. Among the 2459 genes assessed (Appendix A), 384 genes harbored NSNVs across all ST8 isolates. Of these, 129 genes exhibited ≥5% higher variant frequency in ST8 SSTI isolates relative to colonizing counterparts. The top twenty-five differentiating genes are listed in Appendix A and include several associated with metabolism (e.g., transaldolase), membrane function, and phage biology. Notably, numerous genes encoding phage-associated or hypothetical phage proteins, such as Panton-Valentine Leukocidin, bacteriophage integrase, and portal genes, were more frequently variant in SSTI isolates, suggesting an enrichment of mobile genetic elements in invasive strains. Pathway and term enrichment analysis of these 129 genes (Appendix A) revealed significant overrepresentation in KEGG metabolic pathways (sao01100) and functional categories including membrane-associated proteins (GO:0016020), secreted and extracellular proteins (KW-0964, GO:0005576), and cell wall remodeling processes such as peptidoglycan biosynthesis (GO:0009252), regulation of cell shape (GO:0008360), and cell wall organization (GO:0071555). Additionally, genes annotated with DNA binding (GO:0003677) functions were also enriched, potentially implicating transcriptional regulation in adaptation to host tissue environments. These findings highlight ST8-specific variant genes associated with membrane architecture, phage dynamics, and metabolic flexibility that may contribute to the enhanced fitness of SSTI-associated strains in tissue-invasive contexts.

Taken together, these results indicate that NSNVs in *S. aureus* clinical isolates are not randomly distributed but instead cluster in functionally significant pathways. The enrichment of proteolysis, cell wall biogenesis, and metal transport suggests that these variants may confer selective advantages in either colonization (nasal niche adaptation) or infection (tissue invasion and immune evasion). Notably, the overlap between unitig-enriched regions and NSNV-containing genes supports the hypothesis that these genetic changes have functional consequences, potentially influencing strain-specific virulence and persistence.

#### 3.5.4. Distinct Enrichment of Unitigs and NSNVs for Infection vs. Colonization

To better understand whether nonsynonymous nucleotide variants (NSNVs) in SSTI-associated strains reflect functional alterations, we annotated affected genes for conserved protein domains, enzymatic function, and biological pathway involvement. Notably, several NSNV-enriched genes contained domains linked to enzymatic activity or host interaction, including α/β-hydrolase folds, membrane transporters, and components of nucleotide biosynthesis. These functional annotations help contextualize the potential phenotypic consequences of these genetic variants.

Among the 264 genes containing both significant NSNVs and unitig-based associations, several were associated with functions related to nutrient transport, metabolism, and virulence in the context of SSTI. A consistent enrichment of genes involved in metal ion uptake was observed, including *cntB* (*SAOUHSC_02766*), *nikD* (*SAOUHSC_01378*), and *nikA* (*SAOUHSC_00201*), which encode cobalt/zinc/nickel and nickel-specific ABC transporter components and contained NSNVs p.Met16Val, p.Ser15Cys, and p.Met299Thr, respectively. Additional transport-related mutations were observed in an ABC-type phosphate transporter gene (*SAOUHSC_02699*, p.Met8Val) and in *SAOUHSC_02298* (p.Lys23Asn), an uncharacterized ABC transporter predicted to support amino acid uptake. These results point to selection on transporter systems with potential relevance to nutrient acquisition under host-imposed limitations.

Multiple genes associated with lipid metabolism and hydrolytic activity contained mutations significantly associated with SSTI. Lipase 1 (*SAOUHSC_03006*) and Lipase 2 (*SAOUHSC_00492*) harbored NSNVs (p.Leu674Val and p.Asp111Gly, respectively), alongside a BD-FAE family esterase-like gene (*SAOUHSC_00301*, p.Ala31Val). These enzymes are putatively involved in lipid degradation. The presence of NSNVs in Lipase 1 and Lipase 2 may reflect adaptation to promote the activity of additional lipolytic systems, including a distinct α/β-hydrolase gene (*SAOUHSC_02787*), which is not identified within these results but is described in a subsequent section of the study as being part of a candidate pathoadaptive genomic island involved in wound colonization. *SAOUHSC_02448*, another α/β-hydrolase gene with a p.Ile13Ser substitution, also displayed convergent signals of genomic variation.

Several genes encoding immune-modulating factors or cytolytic toxins were among those with significant associations. Two adjacent leukocidin-like genes (*SAOUHSC_02243* and *SAOUHSC_02241*) carried mutations p.Leu322Phe and p.Lys308Gln, respectively. A staphylococcal superantigen-like protein gene (*SAOUHSC_00383*, p.Ile50Val) and the gamma-hemolysin subunit *hlgB* (*SAOUHSC_02710*, p.Gly22Asn) were also among those carrying NSNVs associated with the infection phenotype.

Key components of energy metabolism and biosynthetic processes were also represented. *SirB* (*SAOUHSC_02685*), encoding a sirohydrochlorin chelatase involved in the synthesis of siroheme (a cofactor in nitrite and sulfite reductases), harbored a p.Ile239Thr mutation. *ArgJ* (*SAOUHSC_00148*), involved in arginine biosynthesis, contained a p.Thr188Ala substitution. These findings support genomic variation in nitrogen assimilation pathways. Multiple genes tied to anaerobic respiration and redox processes also carried NSNVs. *NirD* (*SAOUHSC_00949*), the small subunit of NAD(P)H-dependent nitrite reductase, carried a mutation (p.Pro71Gln), potentially indicating selection on nitrate/nitrite reduction pathways under hypoxic conditions. A molybdopterin-binding oxidoreductase (*SAOUHSC_00853*) encoded a p.Val309Leu substitution, further supporting alterations in anaerobic energy production. Additionally, a quinone-binding monooxygenase (*SAOUHSC_00078*, p.Pro17Leu) and a NADH oxidoreductase-like gene (*SAOUHSC_00545*, p.Gly120Val) were also affected.

Components of menaquinone biosynthesis were represented by mutations in *SAOUHSC_02533* (putative *menA* homolog, p.Val125Ala), potentially altering isoprenoid side-chain biosynthesis essential for anaerobic electron transport. Additional redox-related enzymes included *SAOUHSC_01064* (a 4Fe–4S ferredoxin with p.Cys47Arg) and *SAOUHSC_01591* (predicted iron–sulfur cluster protein, p.Pro26Gln), highlighting broader alterations to oxidative stress handling and electron flow mechanisms.

A mutation was also present in *secA2* (*SAOUHSC_02985*, p.Gln308Glu), a component of a paralogous secretion system for specialized substrates, including virulence-associated proteins. A metalloprotease (*SAOUHSC_02971*, p.Ser331Asn) and an uncharacterized M23 family peptidase (*SAOUHSC_00837*, p.Thr112Ala) also carried NSNVs, implicating proteolytic processing enzymes in adaptation to infection.

A distinct subset of genes exhibited non-synonymous mutations exclusively within the colonization-associated genomes, suggesting that these loci may be conserved in SSTI strains but diverged or disrupted in nasal-colonizing isolates. Notably, multiple hypothetical proteins and conserved domains of unknown function were affected, including *SAOUHSC_02057* (p.Glu243Asp), *SAOUHSC_02129* (p.Thr222Ala), and *SAOUHSC_02850* (p.Ala100Gly), indicating potential loss or functional attenuation of uncharacterized elements during niche specialization. Several affected genes are involved in fundamental cellular processes, such as *SAOUHSC_01380* (a predicted acetyltransferase, p.Pro69Gln) and *SAOUHSC_02645* (a cell division-associated protein, p.Glu231Lys), which may be under purifying selection in SSTI strains to maintain virulence or environmental fitness. The observed mutations in *SAOUHSC_00680*, encoding a putative NADH: flavin oxidoreductase (p.Leu188Phe), and *SAOUHSC_01851*, a putative thiamine biosynthesis enzyme (p.Arg21His), suggest differential metabolic requirements between colonizing and infecting strains. Additionally, *SAOUHSC_02687*, annotated as a putative formate/nitrite transporter (p.Val200Ile), exhibited colonization-specific mutation, highlighting distinctions in nitrogen or anaerobic respiration pathways that may be selectively retained in SSTI genomes. The absence of these mutations in SSTI isolates implies selective pressure to preserve intact redox, metabolic, and transport functions that may be critical for survival and persistence in invasive environments.

These findings reveal multiple loci under selection in SSTI-associated *S. aureus*, spanning metal ion transport, lipid hydrolysis, redox and nitrogen metabolism, and virulence factor secretion. The absence of mutations in genes such as *SAOUHSC_02787*, described later in the manuscript (Section 3.7) as part of a lipase-rich α/β-hydrolase island, suggests alternative mechanisms such as gene presence/absence variation or regulatory shifts may contribute to its role in pathogenesis.

#### 3.5.5. Standardization of Unitig Effects and Z-Score Enrichment of Significant Genes

To prioritize functional relevance, z-score transformation of unitig and NSNV data was performed across all gene associations. A subset of unitigs exhibited z-scores > 2, indicating pronounced associations with the outcome variable (Figure 4, whisker plots). NSNVs also displayed z-scores in the extreme ranges, suggestive of genes with disproportionate allele frequency differences between phenotypes (Figure 4, bar plots).

Pathway-level analysis of z-score-ranked unitigs showed significant enrichment in biological processes relevant to host–pathogen interaction. Processes such as cytolysis, transformation competence, cell adhesion, and amino acid metabolism were enriched for high-ranking unitigs. For instance, the process “cytolysis in another organism” demonstrated an average unitig odds ratio of 2.20, with contributing genes including *lukD*, *hlgA*, and *lukF-PV*. Enrichment of unitigs in peptidoglycan catabolism and cell wall organization was observed within CC8, underscoring lineage-specific adaptation signatures. Enriched genes included prophage autolysins (*lytM*, *lytS*), as well as cell wall biosynthesis genes such as *atl*, *murA–G*, *femA/B*, and *tagX*.

In contrast, NSNV enrichment patterns diverged from those observed in unitig analysis. Among the fifty-nine biological processes enriched for unitigs, nine processes showed significantly greater NSNV enrichment in SSTI strains despite unitig enrichment in colonizing strains. These included critical metabolic and stress-adaptive pathways such as menaquinone, isoleucine, and valine biosynthesis, nitrate assimilation, amino acid transport, DNA repair, and SOS response. These were classified as “Essential SSTI Adaptations” due to their importance for in vivo persistence. Conversely, seven processes demonstrated increased unitig associations with SSTI but higher NSNV enrichment in colonizing strains, including pathways tied to DNA processing, transcriptional regulation, and stress responses. This inverse pattern suggests that while certain functions may be expanded or maintained at the structural level in infection isolates, colonizing strains may harbor sequence-level variation in these same pathways, potentially modulating their expression or activity.

#### 3.5.6. Co-Localized Nitrogen Metabolism and Phage Genes Associated with SSTI Strains

Variant analysis identified significant enrichment of nitrogen assimilation and purine/pyrimidine metabolism pathways in SSTI-associated *S. aureus* strains. Genes involved in nitrate assimilation exhibited high odds ratios, particularly *narI* and *nirB*, with average NSNV ORs of 12.35 in SSTI samples. Co-occurrence of these genes was more frequent in SSTI samples: *nirB* and *narH* were present in 89% and 78% of SSTI samples, respectively, versus 35% and 22% in colonization samples. Corresponding odds ratios were 5.22 (*p* = 0.0021) for *nirB* and 4.31 (*p* = 0.0011) for *narH*. Ten nitrogen-related genes (*nirB*, *nirD*, *cobA*, *narZ*, *narH*, *narJ*, *narI*, *nreA*, *nreB*, *nreC*) were detected in a nitrogen-cluster genomic region in 84% of SSTI strains, compared to 27% of colonization strains. These genes frequently co-localized within the same contig in SSTI samples, with an average intergenic distance of 850–1250 bp. In contrast, colonization samples exhibited sporadic distribution and significantly larger intergenic distances, averaging ~3000 bp. Genes frequently co-localizing with the nitrogen-cluster included *sirB* (OR = 5.96, *p* = 9.30 × 10^−5^), *cysG* (OR = 5.22, *p* = 0.0021), and *narH*. The immunoglobulin-binding gene *Sbi* was detected in 55% of SSTI samples and 15% of colonization samples (OR = 4.54, *p* = 0.00033). *narI* and *nirB* co-occurred with other metabolic and anaerobic energy-related genes in 60% of SSTI samples versus 18% of colonization samples (combined OR = 4.88, *p* < 0.001). Spatial analysis showed that in SSTI strains, *nirB*, *cobA*, *narZ*, and *nreA* were located closer together (average distance = 950 bp) compared to colonization strains (average = 3100 bp). Genes such as *fnbB*, *tcaA*, and *gltP* were more prevalent in colonization samples (72%) compared to SSTI samples (25%). For example, *tcaA* was associated with a significantly lower odds ratio in SSTI (OR = 0.25, *p* = 0.0019).

Whole-genome CNV analysis identified 1048 genes with significant differences between phenotypes (110 infection vs. 938 colonize). Of the 110 infection genes, *nifR3* had the highest positive Cohen’s d (1.63, FDR = 0.00013), consistent with CNV profiling results which exhibited elevated copy number in SSTI. This was followed by *SAOUHSC_00397* (DNA-methyltransferase; d = 1.62, FDR = 1.56 × 10^−6^), and *SAOUHSC_00391* (Superantigen-like gene; d = 1.36, FDR = 0.0027). Additional genes with large positive effect sizes included *SAOUHSC_02047* (Phage head morphogenesis gene) and *SAOUHSC_01515* (Autolysin PH gene). Genes with negative effect sizes had higher CNVs in colonization samples. *SAOUHSC_00786* (*nifR3* paralog) had a Cohen’s d of −1.34 (FDR = 0.00036), while *agrC* had d = −1.28 (FDR = 8.65 × 10^−6^) and *lexA* (repressor) had the most extreme negative value (d = −1.96, FDR = 0.0040). Cohen’s d values represent standardized differences in normalized copy numbers between groups, where positive values indicate enrichment in SSTI isolates and negative values indicate enrichment in colonization strains. *nifR3* CNV was elevated in SSTI samples (mean CNV = 1.55) compared to colonization (mean = 0.89), Cohen’s d = −1.35 (*p* = 6.6 × 10^−6^). In 78% of SSTI samples, *nifR3* co-localized with DNA repair and stress-response genes (*recA*, *radC*, *uvrA*) within an average distance of 1200 bp.

In colonization samples, this co-localization occurred in only 12%. Across 74% of SSTI samples, *nifR3* was co-localized with phage elements. Phage proximity was also observed near nitrogen-related genes and genes like *Sbi* and DUF1413-domain proteins. In SSTI strains, anaerobic energy pathways were significantly enriched. The menaquinone biosynthesis pathway had a high NSNV odds ratio of 1.74. Additional enrichment was observed in genes related to the TCA cycle and gluconeogenesis.

Prophage-encoded genes were also detected within pathways involved in anaerobic metabolism and DNA maintenance. Enrichment of restriction-modification (RM) system genes was observed in SSTI strains, including *hsdM* and *mcrC*, both involved in maintaining genome stability. These genes were frequently detected alongside nitrogen metabolism and nucleotide synthesis genes in SSTI samples.

### 3.6. Machine Learning Identifies purL and Branched-Chain Amino Acid Pathways as Key Infection Predictors

Metabolic-centric, unitig-based machine learning analysis of all SSTI and colonizing *S. aureus* clinical samples and was performed. To reduce feature dimensionality and focus on metabolically relevant variation, only unitigs mapping to genes annotated in metabolic pathways (KEGG level 2) were retained for training the model, as determined by DAVID functional annotation analysis. This achieved an accuracy of 0.787 on the test set, indicating that the model correctly classified approximately 79% of the samples. The ROC AUC was 0.8598. This suggests that the model performed well in distinguishing between the two phenotypes (SSTI vs. colonization).

The top twenty most important features were plotted based on their F-scores, and SHAP dependence plots were generated for the top features (Figure 5). Among the top predictive features, purine biosynthesis genes previously identified in unitig-based GWAS were reaffirmed, indicating a consistent metabolic signal across analytical methods. The top three features identified by the model based on their F-scores included, (1) *purL* (F = 21): This feature was the most frequently used in decision splits, indicating a significant role in predicting SSTI; (2) Aspartyl/glutamyl-tRNA amidotransferase subunit B (F = 15): This feature was highly important, frequently contributing to the model’s decisions; and (3) 1-(5-phosphoribosyl)-5-[(5-phosphoribosylamino)methylideneamino] imidazole-4-carboxamide isomerase (F = 15): This feature ranked third in importance, playing a key role in the model’s predictions (Figure 5).

KEGG pathway analysis highlighted several additional significant pathways associated with the top 20 features including (1) Biosynthesis of Amino Acids (KEGG Pathway: sao01230): This pathway was the most enriched, with 52.63% of the top features mapping to genes involved in amino acid biosynthesis (*p*-value = 1.97 × 10^−5^). Genes such as Q2FYR9, Q2G122, Q2FWK1, Q2FZS6, Q2FYH0, Q2FUU2, Q2FY32, Q2FYV3, Q2FXE8, and Q2FW49 were involved in this pathway, indicating a significant role in the metabolic processes of *S. aureus*; (2) Biosynthesis of Secondary Metabolites (KEGG Pathway: sao01110): This pathway was enriched with 73.68% of the top features (*p*-value = 6.1 × 10^−5^). Genes such as Q2FZJ0, Q2FYR9, Q2FWK1, Q2FZS6, Q2FYH0, Q2FV23, Q2FY32, Q2FYV3, Q2FW49, Q2G122, Q2FUU2, Q2FXE8, Q2FXF9, and Q2G1Z6 were associated with the biosynthesis of secondary metabolites; and (3) Valine, Leucine, and Isoleucine Biosynthesis (KEGG Pathway: sao00290): This pathway, related to branched-chain amino acid biosynthesis, was represented by 15.79% of the top features (*p*-value = 0.021). Genes Q2FWK1, Q2FZS6, and Q2FW49 were involved in this pathway, highlighting their potential contribution to essential metabolic functions.

GO biological process analysis further emphasized the involvement of these features in key metabolic processes, including (1) Amino Acid Biosynthetic Process (GO:0008652): Four genes (Q2FWK1, Q2FZS6, Q2FYH0, Q2FY32) were involved in amino acid biosynthesis, with a significant enrichment (*p*-value = 0.00128). This finding aligns with the KEGG pathway results, reinforcing the importance of amino acid metabolism in *S. aureus*; and (2) L-Leucine Biosynthetic Process (GO:0009098): Two genes (Q2FWK1, Q2FZS6) were specifically involved in L-leucine biosynthesis (*p*-value = 0.07468). Leucine biosynthesis is crucial for bacterial growth and survival, particularly in nutrient-limited environments. In terms of molecular function, the analysis revealed: (1) Metal Ion Binding (GO:0046872): Three genes (Q2FYR9, Q2FWK1, Q2FYH0) were involved in metal ion binding (*p*-value = 0.502). Metal ion binding is essential for enzymatic activity and regulatory processes in bacteria; and (2) Magnesium Ion Binding (GO:0000287): Four genes (Q2FZJ0, Q2FY32, Q2FW49, Q2FYB2) were enriched for magnesium ion binding (*p*-value = 0.0265). Magnesium is a critical cofactor for various enzymes and plays a role in stabilizing ribosomes and membranes.

### 3.7. A Conserved α/β-Hydrolase Island Correlates with Increased Clinical Severity in SSTI

To identify the top unitigs which were associated with higher clinical severity and had the least amount of lineage effects, we used a custom python script to first identify unitigs which mapped to a contig and ranked them by their mappability quality (uniqueness of the sequence). This resulted in 129,119/382,883 unitigs, of which 46,931 were significantly associated with infection. We then applied a custom index which filtered for unitigs which had the highest discernment between phenotypes, controlled for lineage effects, and calculated the average PUS score between samples with and without each unitig. PUS (Purulence Ulcer Skin) score is a composite clinical metric derived from wound characteristics, erythema, and patient-reported symptoms, scaled from 0 (colonization) to 15 (most severe infection); the mean PUS score was 2.98 (SD ± 2.97). This resulted in 91 significant (FDR-adjusted) unitigs, of which 66 had mappability scores > 30.

The top unitigs associated with higher PUS scoring mapped to a genomic region encoding proteins with an α/β-hydrolase domain (Table 1, Figure 6), a functionally diverse superfamily encompassing lipases, esterases, and virulence-associated hydrolases (4). While automated annotation labeled these genes as ‘Lipase,’ secondary sequence analysis (InterProScan) indicated broader potential roles in host–pathogen interactions or metabolic adaptation. Adjacent genes were also top unitigs and included lipoproteins and cytosolic proteins, which are frequently implicated in Staphylococcal virulence.

In addition, other top-scoring unitigs mapped to genes within several shared biological clusters including amino acid transport and metabolism (COG:E; *aroA*, *metF*), DNA-binding transcriptional regulation (COG:K; *lysR*, *purR*, *mngR*), plasma membrane (GO:0005886; *ebh*—extracellular binding matrix), ATP-binding (GO:0005524; *ileS*, *cshB*, *ispE*), and zinc ion binding (GO:0008270; *hutl*, *ileS*). The identification of these gene clusters supports our central hypothesis that specialized metabolic and regulatory pathways contribute to the adaptability and virulence of *S. aureus* in purulent infections. Understanding these functional gene clusters could inform future therapeutic strategies by targeting essential metabolic and virulence pathways critical for the successful colonization and infection of the host.

We also developed interpretation thresholds (Appendix A) and ran an independent bGWAS with PUS scores (Appendix A) which identified 15 additional significant unitig-mapped genes with a distribution of PUS score effect sizes from 1.5 to 3.1, with a mean of 2.12 and median of 2.04 (Appendix A). Several of the top hits included phage-associated autolysins and holins (e.g., ABD31075-ABD31076; *lytO*-Holin), ABD30597-ABD30598 (autolysin-Holin), each with PUS score effect sizes above 2.8, suggesting potential roles in bacterial cell wall degradation and interbacterial antagonism. Notably, ABD30607 and ABD31226, both encoding lysostaphin-like peptidoglycan hydrolases, also demonstrated high effect sizes (2.5 and 2.1, respectively), consistent with mechanisms involved in biofilm remodeling or lysis-mediated competition. Additional hits included surface proteins (ABD31801; *sasG*) linked to biofilm formation, penicillin-binding protein 3 (ABD30728), and several hypothetical phage proteins and anti-repressors, highlighting diverse mechanisms potentially contributing to increased purulence.

### 3.8. Distinct Prophage Cargo and Gene Products Associated with Infection Severity

Given the distinct genomic differences in phage-related regions between SSTI and nasal colonization strains (Appendix A), we conducted a comprehensive viral DNA screening of all 157 genome assemblies using VirHunter (v1.0). The median number of viral fragments per genome was similar between phenotypes, specifically 1441 in SSTI isolates and 1419 in colonizing isolates. Despite similar abundance of predicted viral fragments between phenotypes, we hypothesized that there were distinct taxonomic distributions and functional profiles between SSTI and nasal colonization strains.

Phage region annotations across the isolates were analyzed for taxonomic distribution using lowest-level classification data from Phastest. Counts were normalized by sample size within SSTI (*n* = 126) and colonization (*n* = 31) groups to control for group imbalance. Normalized bar plots of the lowest taxonomic assignments revealed that several phage lineages were evenly distributed, whereas others were more prevalent in SSTI-associated genomes (Appendix A). Specifically, a four times higher count (sample size normalized) of *Triavirus* taxa regions were present in SSTI vs. nasal colonization strains. Abundant counts of *Biseptimavirus* and *Dubowvirus* were also present in both phenotypes along with *Peeveelvirus* and *Phietavirus* representing slightly higher counts in nasal colonization strains.

A total of 30 distinct phage gene annotations showed statistically significant differences (*p* < 0.05) between groups, with odds ratios (OR) ranging from 0.05 to 461.70 (Appendix A). Among the most enriched in SSTI isolates were ABC transporter (OR = 461.70, *p* = 1.24 × 10^−20^), DNA binding protein (OR = 324.68, *p* = 1.52 × 10^−18^), enterotoxin type A/*speL* (OR = 303.42, *p* = 3.54 × 10^−18^), and acetyltransferase (OR = 305.34, *p* = 6.76 × 10^−17^). Genes involved in DNA replication and repair such as DNA primase, DNA helicase, endonuclease, DNA polymerase, and Gp2.5-like ssDNA binding protein were also enriched in SSTI genomes. In contrast, genes such as Mu Gam-like end protection, Lar-like restriction alleviation protein, and esterase/lipase were more prevalent in colonizing isolates.

To further resolve associations between phage content and SSTI, we screened all isolate genomes against a curated phage protein database using BLAST, applying a 90% amino acid identity threshold. We identified several proteins derived from the top Staphylococcus phage taxa phiIPLA35 [Triavirus] that were disproportionately enriched in SSTI isolates. Specifically, Gp5 (a metallo-protease), Gp41 (a major capsid protein), and Gp11 (an IS630 family transposase) showed the highest differential prevalence in SSTI versus colonization isolates (Appendix A). These proteins were detected in ≥25% more SSTI genomes than colonization genomes, suggesting a functional link between these phage elements and the infection phenotype.

To investigate whether these phiIPLA35 gene products were associated with clinical markers of disease severity, we analyzed their relationships with wound size, a proxy for SSTI severity. Across all SSTI patients, the median wound size was 4 cm, with a maximum of 17 cm. Using bootstrapping with 1000 resamples, we computed the mean difference in wound size between isolates carrying each phiIPLA35 gene product and those that did not. Confidence intervals were used to assess statistical significance (Appendix A). Presence of Gp5, Gp35, and Gp41 were associated with significantly larger wound sizes. In contrast, presence of Gp49, Gp7, Gp12, and Gp50 were associated with significantly smaller wound sizes, suggesting a potential protective effect.

The top phiIPLA35 gene product linked to SSTI, Gp5, is a metallo-endopeptidase of the ImmA/IrrE family [15]. Multiple alignment of Gp5 with homologs across species revealed structural conservation with similar virulence factors in other bacteria, including *Clostridioides difficile* (Appendix A). Gp35, a RinA family transcriptional activator, and Gp41, a major capsid protein belonging to the Gp5 superfamily, were also strongly associated with cellulitis [16]. Together, these data suggest that prophage-encoded virulence, regulatory, and structural proteins may contribute to the observed variability in SSTI pathogenesis and wound severity, independent of total phage burden.

### 3.9. Metagenome-Assembled Genomes from Colonized Nasal Swabs

To evaluate the environmental predisposition in colonizing strains, we performed exploratory analyses of the nasal compartment. We performed nasal metagenomic assembly, applying conventional “high-quality” criteria (e.g., completeness ≥90% and low contamination; see Appendix A). This yielded seven well-resolved bins dominated by multiple *Staphylococcus* species, *Enterococcus*, and *Micrococcus*, with completeness above 84% (several exceeding 95%) and minimal contamination (Appendix A). The integrated summary table combines taxonomic/quality metadata with functional annotations from core metabolism (GO), accessory gene families (COG, EC), and virulence factor profiling (VFDB), enabling a unified view of retained essential biology alongside potential pathogenicity signatures (Figure 7).

All high-quality bins uniformly retained the defined set of 94 core GO biological process terms, encompassing amino acid biosynthesis, genome maintenance and repair, central carbon and energy metabolism, nucleotide metabolism, transcription/RNA processing, translation/protein maturation, transport, stress response, and cell envelope/division processes (Figure 7B). In *Staphylococcus* bins, this conserved metabolic backbone was paired with variable accessory gene capacity, including COG and EC profiles (Figure 7C) consistent with environmental sensing, nutrient acquisition, and interactions with host-derived substrates. Detected enzymatic activities span cofactor metabolism, energy production, and substrate interconversion, indicating active participation in diverse biochemical conversions relevant to persistence in the nasal niche. *Enterococcus* exhibited the largest accessory repertoire by gauging COGs and numerous EC-numbered enzymatic activities suggesting high metabolic flexibility, while *Micrococcus* maintained a smaller, diverse accessory set (Figure 7C).

Virulence-associated genes (Figure 7D) harbored within the *Enterococcus* bin included components of the cytolysin system (e.g., *cylA*, *cylB*, *cylI*, *cylM*, and regulator *cylR1*), adhesion and aggregation machinery (e.g., *efaA*, aggregation substance EF0149), and immune modulation factors (capsule-related enzymes such as *cpsA*/*cdsA*). *Micrococcus* showed sparse virulence factor annotation, aligning with its common commensal status. These factors appear in combinations that could promote host adherence, biofilm formation, immune evasion, and cytotoxicity—characterizing a flexible opportunist capable of transitioning toward pathogenic interactions under favorable conditions.

Beyond these bin-level patterns, several infection-enriched loci detected in the *S. aureus* SSTI isolates were not recovered in the nasal *Staphylococcus* metagenomic bins. Notably, lipolytic enzymes—including Lip2 and a conserved α/β-hydrolase island linked to higher clinical severity and lipid nutrient acquisition. In the purine axis, the salvage enzyme *deoC* (deoxyribose-phosphate aldolase) was not observed in colonizing metagenomic bins. Additionally, the menaquinone pathway (e.g., *menA*/*menB*) were only partially represented in the bins. Finally, nitrogen assimilation/respiration modules (e.g., co-localized *nirB*/*narH* cluster) were not captured by the metagenomic bins. Together, the missing loci underscore pathoadaptive remodeling toward energy generation and nutrient scavenging during SSTI that are lacking in nasal MAGs.

## 4. Discussion

Our genomic analysis of *S. aureus* isolates from SSTIs, and asymptomatic colonization reveals a coordinated interplay between lineage-specific genomic architecture, metabolic adaptations, and mobile genetic elements that potentially drives the transition from commensalism to pathogenicity. Herein, we identify: (1) a clonal complex distribution skewed toward CC8 in infections, (2) metabolic pathways—particularly nitrogen utilization and nucleotide biosynthesis—linked to prophage-mediated virulence enhancement, and (3) a conserved surface-associated α/β-hydrolase locus with niche-specific structural integrity.

The overrepresentation of CC8 among SSTI isolates (68% vs. 10% in colonizers) and its association with distinct genomic features suggests this lineage might have evolved specialized mechanisms for tissue invasion and survival in infection microenvironments. These findings provide evidence that strain-intrinsic factors, in addition to host factors, significantly influence pathogenic potential.

Conserved nitrogen metabolism gene cluster (*nirB-narH-narJ*) exhibited structural and functional specialization in SSTI strains. These genes were more frequently co-localized in infection isolates and displayed reduced intergenic distances, consistent with coordinated expression and possible selective retention of this cluster. Corresponding increases in copy number, particularly for *nifR3*, and co-localization with DNA repair and phage-associated genes highlight a genomic niche of coordinated regulation for redox metabolism and genome stability. Prophages may thus serve dual roles in *S. aureus* adaptation—acting as vehicles for horizontal gene transfer while simultaneously encoding functional traits that directly support bacterial survival. Similar adaptation was observed in *Escherichia coli*, where nitrate reductase systems regulate biofilm formation and intracellular persistence, and phage-mediated regulation of nitrogen metabolism has been documented in multiple bacterial pathogens [17,18].

The enrichment of cytolysis, cell wall metabolism, and cell adhesion pathways in unitig-based analysis aligns with known virulence strategies employed by *S. aureus*, including leukocidin production, immune evasion, and host tissue destruction [19]. Notably, divergent enrichment patterns between unitigs and NSNVs in the same pathways suggest distinct evolutionary pressures shaping infection versus colonization phenotypes. NSNVs were more frequently enriched in amino acid biosynthesis (e.g., isoleucine, valine), nitrate assimilation, and stress response pathways in SSTI isolates, reflecting sequence-level adaptation to the inflammatory, hypoxic, and nutrient-limited conditions characteristic of abscess environments. This divergence underscores complementary roles for variant types in bacterial adaptation, with structural variations supporting rapid genomic innovation, and point mutations fine-tuning protein function [20,21].

Metabolic remodeling in SSTI-associated strains extended beyond nitrogen assimilation to include strong evidence supporting purine biosynthesis, thiamine metabolism, and menaquinone pathway genes. The convergence of both GWAS and machine learning results on purine biosynthesis genes strengthens the likelihood of these pathways playing a critical role in SSTI pathogenesis. Multiple purine biosynthetic genes (*purL*, *purK*, *purM*, *purF*) were among the top predictors in machine learning classification and showed high allelic diversity, suggesting active selection for variants that promote replication, intracellular persistence, or antibiotic tolerance. These findings align with previous reports linking purine metabolism to stress resistance and virulence regulator expression [22,23,24,25]. In particular, the transcriptional regulator *purR*, which modulates purine synthesis, is known to intersect with global regulators such as *sarA* and *sigB*, and its derepression has been associated with increased virulence in endocarditis and intracellular infection models [24,25].

Our results also reveal enrichment of non-synonymous mutations in menaquinone biosynthesis (e.g., *menA*, *menB*) and the TCA cycle, supporting the notion that SSTI strains engage alternative electron transport and carbon flux pathways to generate ATP in oxygen-limited tissues. This metabolic plasticity parallels previous work showing that *S. aureus* can shift between terminal oxidases under hypoxia and that menaquinone is essential for anaerobic respiration [26,27]. The identification of mutations in menaquinone pathway genes may reflect regulatory rewiring or feedback inhibition resistance, potentially enabling persistent growth in abscess microenvironments [28]. From a therapeutic perspective, these adaptations highlight the vulnerability of *S. aureus* respiratory chains to targeted inhibition, especially under host-imposed metabolic stress.

Among virulence-associated loci, α/β-hydrolase domain-containing genes emerged as key contributors to infection severity. Multiple unitigs mapping to this region were associated with increased PUS scores, and while not all members carried NSNVs, their consistent enrichment in SSTI isolates suggests selective retention. Lipase 1 and Lipase 2 harbored NSNVs potentially supporting functional redundancy or cooperative activity with α/β-hydrolase enzymes (e.g., SAOUHSC_02787), which are in a pathoadaptive island. Given the known roles of bacterial α/β-hydrolases in lipid degradation and immune modulation, this region may support nutrient acquisition or inflammatory suppression in vivo [29,30,31]. Adjacent cytosolic and lipoprotein genes also ranked highly, implicating broader regulation of lipid handling and host interaction in virulence. These findings support how the genomic plasticity and metabolic flexibility of *S. aureus* can potentially support transition from commensalism to virulence. Thus, the surface associated α/β-hydrolase locus and specialized nitrogen/anaerobic metabolic pathways could emerge as potential targets for therapeutic intervention and diagnostic development.

Colonization-specific mutations were identified in genes tied to membrane stability, thiamine biosynthesis, and formate/nitrite transport (*SAOUHSC_02687*), suggesting niche adaptation to the nasal mucosa. These loci were conspicuously absent in SSTI strains, implying purifying selection to maintain functional redox and metabolic pathways critical for tissue invasion. The contrast between conserved metabolic regulators in infection and degenerated variants in colonization reflects differential selective pressures exerted by the host across anatomical niches.

The lineage-specific distribution of RM systems further supports this model. CC8 strains were enriched for RM-II, while CC30 favored RM-IV, echoing previous work showing that RM systems function beyond restriction immunity, also influencing transcription, DNA repair, and HGT regulation [32,33,34]. The co-localization of RM genes (*hsdM*, *mcrC*) with phage elements, nitrogen metabolism, and DNA repair genes in SSTI strains suggest an integrated role in maintaining genomic stability under immune and environmental stress.

Genome-wide association and z-score transformation of variant data revealed that genes involved in amino acid metabolism, metal ion binding, and virulence regulation were significantly enriched in SSTI genomes. Machine learning models identified *purL* and multiple amino acid biosynthesis genes as top predictors of infection, and enrichment analysis reinforced the importance of branched-chain amino acid pathways, secondary metabolite biosynthesis, and cofactor production in virulence. These findings support a model of convergent metabolic specialization among invasive *S. aureus* strains.

Phage-derived genes—particularly those from phiIPLA35—were strongly associated with both infection phenotype and wound severity. Proteins such as Gp5 (a metallo-endopeptidase), Gp35 (a transcriptional regulator), and Gp41 (a major capsid protein) were significantly enriched in SSTI isolates and correlated with larger wound size, implicating these elements in enhanced virulence. Conversely, other phiIPLA35 elements were associated with smaller wound size, suggesting a modulatory role in infection severity. These findings support a nuanced model where prophages encode a spectrum of virulence and regulatory functions that can amplify or dampen host damage depending on their context and expression.

Our metagenomic analysis of nasal swabs further contextualizes the transition from commensal colonization to invasive infection by revealing extensive genomic diversity among nasal microbiota, including multiple well-resolved *Staphylococcus* species alongside *Enterococcus* and *Micrococcus*. The retention of robust core metabolic functions, encompassing amino acid biosynthesis, nucleotide metabolism, genome repair, and energy production pathways, underscores a conserved metabolic backbone essential for persistent colonization. Importantly, the distinctive accessory gene repertoires and virulence profiles identified among these nasal taxa, particularly in *Enterococcus* and *Staphylococcus* bins, indicate latent pathogenic potential, which may become activated under altered host immune conditions or environmental shifts. These findings align with previous observations that opportunistic pathogens within commensal niches can rapidly adapt metabolic and virulence pathways to facilitate invasive disease under selective pressures such as immune compromise or microbiome disruption [35,36,37]. Thus, our data reinforces the concept of dynamic host–pathogen interactions within the nasal microbiota and highlights potential commensal reservoirs for *S. aureus* pathogenicity determinants.

This study has limitations. First, while CNVs reflect read depth relative to MLST housekeeping genes, the underlying mechanism—e.g., chromosomal duplications, prophage integration, or plasmid acquisition—cannot be fully resolved with short-read data alone. Furthermore, the nine-year range of SSTI samples may introduce temporal variation in circulating strains and clinical practices. Colonization samples were collected during a shorter window in 2015, which may limit temporal comparability with SSTI isolates. To mitigate confounding, the year of isolate collection was controlled for in our models. Third, in our analysis, lytic versus lysogenic potential was not directly assessed, representing a limitation and opportunity for future prophage induction studies. Fourth, significant GWAS association with increased clinical severity suggests this region may contribute to infection progression, though further mechanistic studies are needed to resolve its precise function.

Our findings point to several potential therapeutic targets that include the disruption of nitrogen assimilation (*nirB*, *narH*, *nifR3*), purine biosynthesis (*purL*), or the menaquinone pathway (*menA*, *menB*). Likewise, targeting phage-regulated hydrolases or transcriptional regulators could reduce virulence without broadly disrupting microbiota.

## 5. Conclusions

This study assessed the underlying drivers of *S. aureus* SSTI pathogenesis, where CC8′s metabolic versatility, phage-mediated gene content, and specialized regulatory networks converge to support invasive infection from commensalism. Metagenomic analyses further revealed core metabolic capacities and distinctive virulence profiles among nasal commensals, emphasizing their latent potential for opportunistic pathogenic transitions. These findings highlight the complexity of host–pathogen–microbiota interactions and the need for targeted therapeutics tailored to the infectious and colonizing niches; they offer actionable insights for combating community-associated *S. aureus* infections.

## Figures and Tables

**Figure 1 microorganisms-13-02023-f001:**
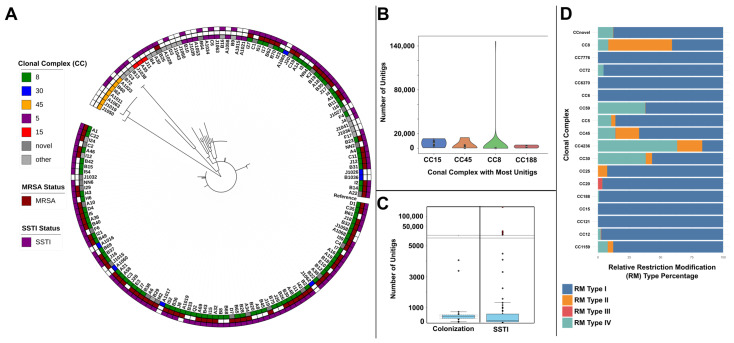
Clinical and molecular characteristics of *Staphylococcus aureus* from skin and soft tissue (SSTI; *n* = 131) and nasal colonization (*n* = 31) in the community. (**A**) Core genome single-nucleotide polymorphic (SNP) tree of 157 community skin and soft tissue (SSTI; *n* = 126) and nasal-colonizing (*n* = 31) *Staphylococcus aureus* isolates and NCTC 8325 (CC8 reference isolate). Metadata provided for methicillin resistance phenotype, SSTI status, and Multilocus Sequence Typing (MLST). Tree branch lengths reflect genetic distances derived from SNP profiles. The segmented inner ring shows MLST (Multilocus Sequence Typing) types, with the following color scheme: green for type 8, blue for type 30, orange for type 45, purple for type 5, red for type 15, gray for novel or untyped isolates, and light gray for other types not specified. The middle metadata ring denotes MRSA (methicillin-resistant *Staphylococcus aureus*) status, where dark red indicates methicillin-resistant samples and white represents methicillin-susceptible samples. The outer ring displays SSTI status, with purple for SSTI and white for nasal-colonizing samples. (**B**) A violin plot of the distribution and density of unitig counts across these clonal complex (CC) groups. The *x*-axis is ordered by the average unitig count for each CC type. Outliers were highlighted and counts above a threshold of 5500 unitigs were displayed on an extended *y*-axis. (**C**) Total count of unitigs per sample grouped by infection phenotype (SSTI or colonization). Black dots represent individual sample unitig counts; red dots indicate outlier samples exceeding the 5500 unitig threshold. (**D**) Distribution of restriction–modification (RM) system types among isolates, shown as stacked bar plots by clonal complex.

**Figure 2 microorganisms-13-02023-f002:**
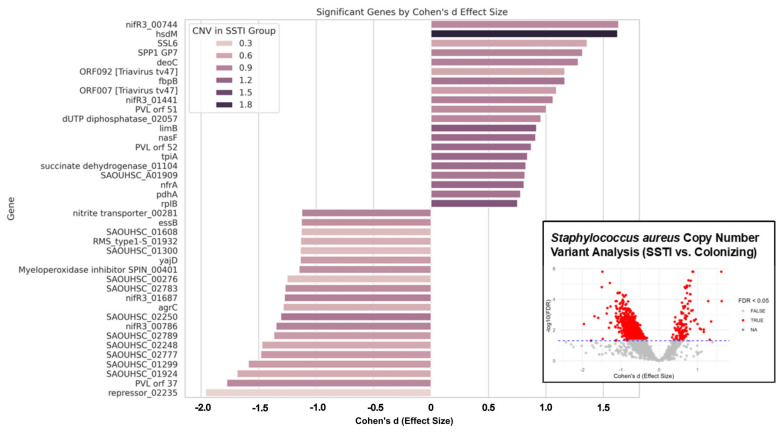
Bar plot of significant copy number variant genes between SSTI and colonization. This plot displays the effect sizes (Cohen’s d) for genes with significant copy number variations (CNVs) between skin and soft tissue infection (SSTI) and colonization groups, filtered by an FDR threshold of ≤0.05. The color gradient represents CNV values in the SSTI group, with each bar labeled by the associated gene name. High Cohen’s d values indicate a stronger effect size, suggesting potential relevance of these genes in distinguishing between infection and colonization phenotypes. NCTC 8325 used as reference.

**Figure 3 microorganisms-13-02023-f003:**
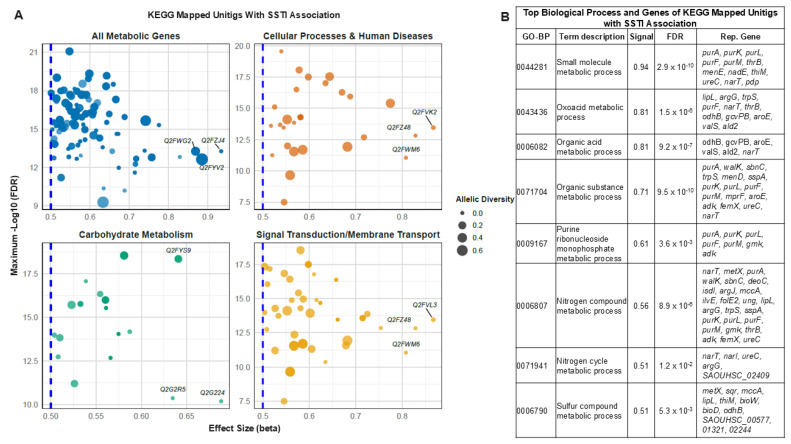
*Staphylococcus aureus* gene −log_10_ lineage-adjusted *p*-value vs. average effect size (skin and soft tissue infection [SSTI]). (**A**) Only genes (Uniprot Accessions) with transformed *p*-value ≥ 6.3 (threshold) and effect size ≥ 0.5 plotted. Dot sizes relate to allelic diversity, which is calculated by subtracting average unitig minor allele frequency (MAF) from average unitig allele frequency (AF). Positive allelic diversity values represent how much more frequent the major allele is compared to the minor allele. Q2FWG2 (*thiM*); Q2FZJ4 (*purK*); Q2FYV2 (*thrB*); Q2FVK2 (*hlgC*); Q2FZ48 (*ftsY*); Q2FWM6 (AgrD); Q2FYS9 (Aconitate hydratase); Q2G2R5 (PTS lactose-specific EIIA); Q2G224 (*deoC*); Q2FVL3 (L-cystine and nickel ABC transporter permease). (**B**) Unitig-mapped genes with an average effect size > 0.5 and lineage-adjusted *p*-value below the threshold and allele frequency > 5% were KEGG-annotated with DAVID.

**Figure 4 microorganisms-13-02023-f004:**
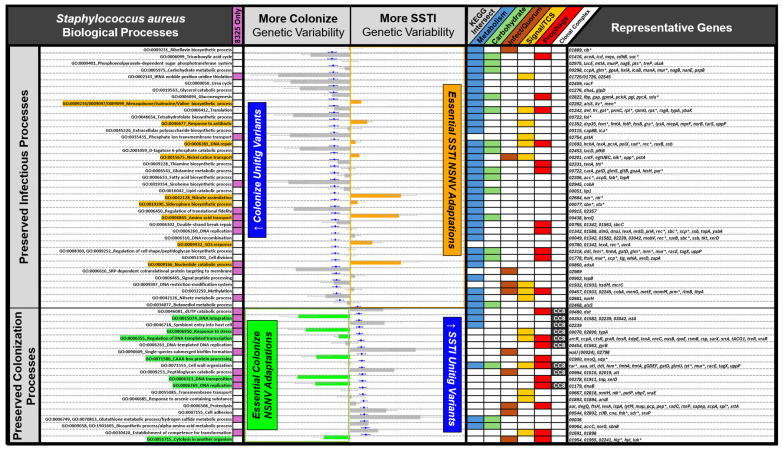
*Staphylococcus aureus* skin and soft tissue vs. nasal colonization genetic variant analyses. Relationships between biological processes, lineage dependence, and associated representative genes across Staphylococcus aureus samples are provided. Each row represents a Gene Ontology Biological Processes term, with associated odds ratios from Z-score transformed unitig counts and log_10_-transformed odds ratio non-synonymous nucleotide variant (NSNV) analyses (NCTC 8325 reference genome). Dashed horizontal line indicates a phenotypic shift in unitig process enrichment. Vertical red dotted lines represent no difference between Kyoto Encyclopedia of Genes and Genomes (KEGG) are highlighted for essential adaptation processes. Metabolic indicates mappings to sao01100 (Metabolic pathways), TCS/Infection indicates mappings to both sao02020 (Two-component system [TCS]) and sao05150 (*Staphylococcus aureus* infection). Clonality refers to clonal complex (CC). Blue represents no association to any specific CC/MLST; Red indicates lineage effects. * Indicates multiple representative genes which share the same prefix. 8325 only = biological processes with NSNV only present when referenced to 8325, not ATCC 27217.

**Figure 5 microorganisms-13-02023-f005:**
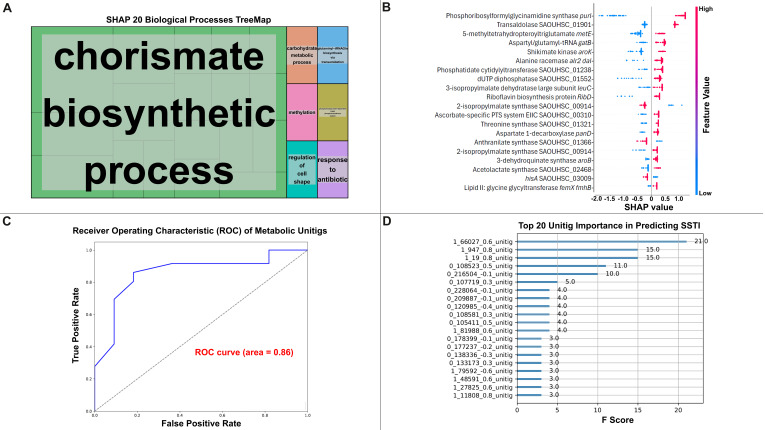
(**A**) SHAP top twenty genes biological process treemap. (**B**) SHAP plot of unitigs with names of mapped gene names. (**C**) ROC of XGBOOST model build by unitigs mapped to metabolic genes from 156 *Staphylococcus aureus* clinical isolates. (**D**) XGBOOST important unitigs which map to metabolic genes for predicting purulent cellulitis. ‘1_19_0.8_unitig’ maps to *purL* (accession: Q2FZJ0); ‘1_947_0.8_unitig’ maps to Aspartyl/glutamyl-tRNA amidotransferase subunit B (accession: Q2FWZ0); and ‘1_66027_0.6_unitig’ maps to 1-(5-phosphoribosyl)-5-[(5-phosphoribosylamino)methylideneamino] imidazole-4-carboxamide isomerase (accession: Q2FUU2).

**Figure 6 microorganisms-13-02023-f006:**
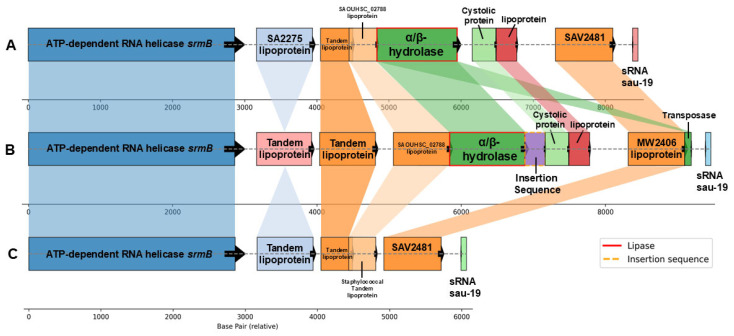
Genomic regions with α/β-hydrolase domain gene from SSTI and Colonizing *Staphylococcus aureus* strains. (**A**) α/β-hydrolase domain gene genomic context from SSTI strain. (**B**) α/β-hydrolase domain gene genomic region from nasal-colonizing strains with insertion sequence disrupted *SAL3*. (**C**) Genomic region of nasal-colonizing strains which lack α/β-hydrolase domain gene. Arrows indicate 5′ to 3′ strand direction across genomic regions. Color links indicate sequence homology between related genomic regions.

**Figure 7 microorganisms-13-02023-f007:**
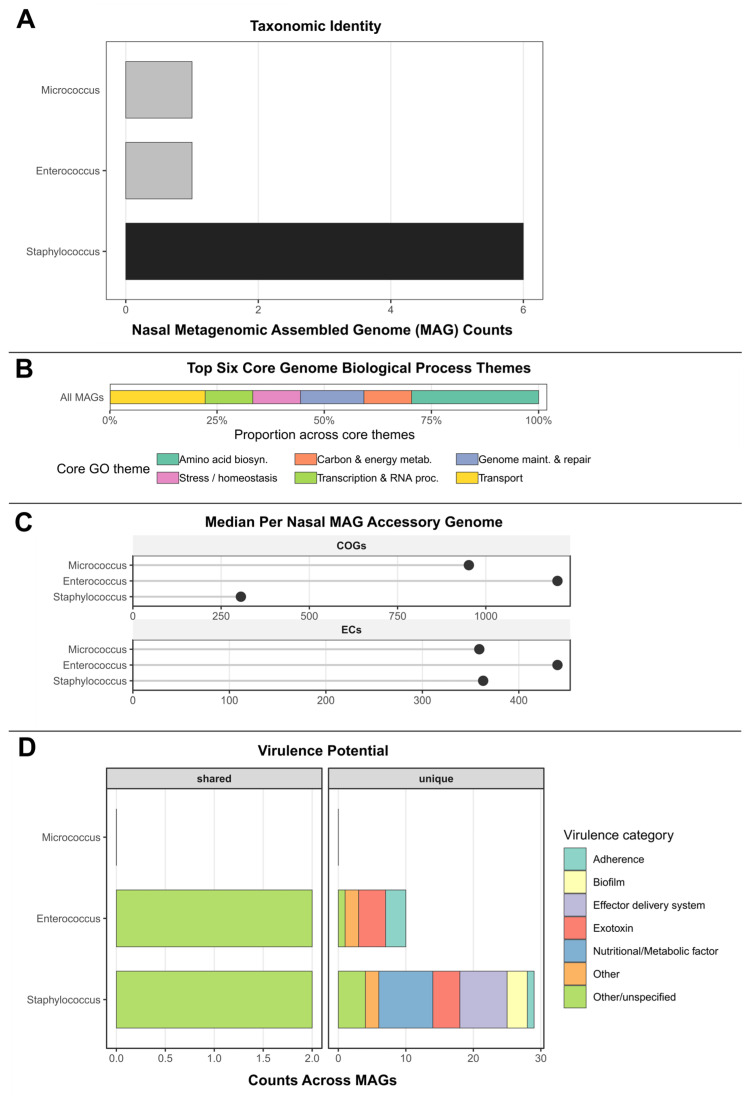
MAGs were taxonomically assigned and functionally annotated. (**A**) Number of MAGs per genus. (**B**) Dataset-wide composition of the top six core GO themes (after excluding “Unassigned/other”). (**C**) Accessory function summarized as the median number of COGs and ECs per MAG for each genus. (**D**) VFDB hits are tallied across MAGs and grouped by category; panels split shared (detected in ≥2 genera within this dataset) vs. unique (detected in only one genus). “Other/unspecified” pools remaining categories. MAG = metagenome-assembled genome; COG = Cluster of Orthologous Groups; EC = Enzyme Commission; VFDB = Virulence Factor DataBase. Counts reflect detections in this dataset and do not imply absence in nature.

**Table 1 microorganisms-13-02023-t001:** Top Unitigs Associated with Clinical Severity.

Unitig-ID	Gene/ Product	AF	*p*-Value ^A^	Beta (SE)	HI	Mapq
p53236_c5_2	α/β-hydrolase	0.89	1.2 × 10^−14^	0.78 (0.09)	0.57	60
p53219_c4_1	α/β-hydrolase	0.89	4.5 × 10^−17^	0.81 (0.08)	0.61	60
p53219_c4_3	α/β-hydrolase	0.89	4.5 × 10^−17^	0.81 (0.08)	0.61	60
p53219_c4_4	α/β-hydrolase	0.89	4.5 × 10^−17^	0.81 (0.08)	0.61	60
p53236_c5_1	Cytosolic protein	0.89	1.2 × 10^−14^	0.78 (0.09)	0.57	60
p53236_c5_3	Cytosolic protein	0.89	1.2 × 10^−14^	0.78 (0.09)	0.57	54
p53216_c2_2	Cytosolic protein, Lipoprotein (spans both)	0.88	3.3 × 10^−15^	0.74 (0.08)	0.58	30
p53219_c4_2	Lipoprotein	0.89	4.5 × 10^−17^	0.81 (0.08)	0.61	60

Abbreviations: AF: allele frequency; HI: heritability index; Mapq: mappability quality; SE: beta standard error. (A) *p*-value is lineage adjusted based on tree distance. Unitig sequences: p53236_c5_2: CCATTAGCGATCAAAGATGTGAGCGGCCTTTATGCCGATCAAGAAGAACTTAAAAAATTGATTGAAAAGTACGATGGTCACATTGTAAGATTTGTGTCTGATGAAGACGAATTAGATGCAGGTGTCCGCAATCATTTATATGAAACTGCTGGAGAAAAAATAGTACTTAAAAATGGAGAAGGCCATGCAATGAGTGGTATTTTAATGAGCAGAACACAGGCTATAATCTTAGCTGAATTAAACAAAGTTAAAGGCTACCAAGACGAAAATAATAAAGCATTAAAATCCGTTCGTAAACAAACGAGGCATAGATTACATAAAGTAGAGACGTTAAGAGCGAATTGGATTCAAA; p53219_c4_1: GAGTGAAACGCTAAAGGATATTGGGGCCGATGTCAATATTGGCCTTCATTCCGTCACAGATAAAGATCCACATTATAAAAATACCCAA; p53219_c4_3: ACTTTATCAAAAATATCAAAAAAGACTATGATATTGATATTATTACCGGACATTCGCTGGGCGGTAGAGATGCGATGATTTTAGGTATGAGTAATGATATTAAACATATCGTTGTGTATAATCCAGC; p53219_c4_4: AAAGTTACTTTAGGTATGACAGGTACTAATGTACACAAAGACGCAATATTAAAACAAACATTTGGTGTTCCTTCTTATCAAGGATATATAGATG; p53236_c5_1: AAAAGAAAAATTGATGATAAGATAAAGAAATTAAATGATGTTTATAAAAATTGTAATGGCTATATCGCAAAAATTAAACAGAGTATCGAAGCAATTGTTTCTAATGACCAAATGTTAGCGAGCCAGATTGATGGGATGATGTAATGTTTACTACGTATAAAAATATTAATGAACTTGAAAATGCCTATGATGAAGAAAGAAA; p53236_c5_3: ATTCGTTTCTCTATCGAAAGATTCTATAATACGTGTCATCCTGATCATAGCGTCTTCACTATAATTCATTTTATGTTTGAG; p53216_c2_2: TAAAATCAAGTCCCACGCTTATTCATCTCCTTCAATTCTGTCAGATTGTTTTAGATATTCTCTTCTTAA; p53219_c4_2: CCATTTGGTCAAGTAATATTTTCTCATCACTTTTCAGCTTATTTTGATGATCATGATGCATACGAACATGTTCAGGCGCAATAACCTCACTAATCTGTTCTTTTTCCTTCATTTGTTCCAATACTGCTAAATCTGGTAAACCA.

## Data Availability

The original data presented in the study have been deposited and are openly available in the NCBI BioProject (PRJNA1246601). All analysis scripts and annotation pipelines are available at: https://github.com/codyadamblack/starnet-infection-gwas (deposited and accessed on 10 April 2025).

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
