# Peer review of "Pathoadapative Genomic Determinants of Staphylococcus aureus Community Skin Infections and Nasal Colonization"

_microorganisms, 2025, doi:10.3390/microorganisms13092023_

Round 1
Reviewer 1 Report
Comments and Suggestions for Authors
The article is outstanding in terms of the scale and complexity of the bioinformatic approaches used, but there remain some issues regarding the interpretation of the data. I believe the authors should explicitly state the study's limitations and be more cautious in their conclusions.
Line 99. What was the coverage? Was the quality of the obtained assemblies assessed in any way? Did the key genes used for further analysis (e.g., spa or genes used in MLST) assemble successfully in all genomes?
Line 107. “Core genome alignments were used to construct a maximum-likelihood phylogenetic tree using FastTree“ Please add details on how you obtained the core genome alignments. Does this refer to a concatenation of all variable positions without gaps, identified using Snippy? Or was a more complex approach used?
Line 294. What mechanistically underlies CNV? Genomic duplications, multiple integrated phages, gene transfer to plasmids, prophage induction in cultures? Can this be described at least for a subset of the mentioned genes, if not for the draft genomes obtained in this study, then perhaps using examples from complete genomes in other studies?
Line 336. “The produced Q-Q plot (Figure A2) lacked lower p-value “shelving” indicating a well-controlled population structure.” Please clarify how did you get this conclusion.
Line 357. “Notably, one SSTI sample (SAMN47787133) and four colonization samples (SAMN47787249, 358 SAMN47787255, SAMN47787237, and SAMN47787245) showed significant enrichment in 359 phenotype-shared unitigs (165,493–172,295)” Can it be explained by phylogenetic relatedness?
Line 387. “indicating strong directional selection for its major allele in infection-associated strains. ” Could this simply be genetic hitchhiking, since these alleles are most prevalent in clonal complexes associated with infection but are not directly under directional selection? What is the difference between the alleles of these and other genes, is there a biological interpretation? This applies not only to this section but to the study as a whole.
Line 397. “indicating increased transcriptional activation” Could you elaborate on how this conclusion was reached?
Line 407. “Environmental information processing pathways were represented by genes related 407 to both signal transduction and membrane transport. hlgC (Q2FVK2), a component of the γ-hemolysin pore-forming toxin complex ” I would not group Environmental information processing and toxins together, as they have entirely different functions, and the toxin is not involved in sensing and adaptation to host-associated metabolic cues.
Line 487. “points to enhanced metabolic self-sufficiency.” Enhanced in which group, nasal colonization or infection?
Line 490. “These findings support the concept that S. aureus strains associated with SSTI exhibit an expanded metabolic repertoire” Expanded or degraded? How was this determined based on NSNVs? This applies not only to this section but is a general question regarding the entire analysis.
Line 653. “Enrichment of unitigs in peptidoglycan catabolism” Prophage autolysins that are silent until prophage induction?
Line 620. “GO 620 Biological Process Terms from top to bottom: GO:0009231, GO:0006099, GO:0009401, GO:0005975…”, Line 630 “The full list of truncated genes names: accB, accC, accD, fabD…” I am certain these lists should not be here.
Author Response
Reviewer #1:
- The article is outstanding in terms of the scale and complexity of the bioinformatic approaches used, but there remain some issues regarding the interpretation of the data. I believe the authors should explicitly state the study's limitations and be more cautious in their conclusions.
Response: We thank Reviewer #1 for the thoughtful and detailed review. We are grateful for the recognition of the manuscript’s scale and complexity and have revised the manuscript to address the valuable feedback provided. We agree and have detailed study limitations in the Discussion section.
- Line 99. What was the coverage? Was the quality of the obtained assemblies assessed in anyway? Did the key genes used for further analysis (e.g., spa or genes used in MLST) assemble successfully in all genomes?
Response: Yes. Assembly quality was evaluated using FastQC. All key loci (e.g., MLST, spa) were successfully assembled. We added Supplemental Data 2 and Supplemental Table 8 which summarizes the FASTQC raw data for every fastq file and delineate the spa, MLST, and SCCmec status of all samples, respectively.
- Line 107. “Core genome alignments were used to construct a maximum-likelihood phylogenetic tree using FastTree“ Please add details on how you obtained the core genome alignments. Does this refer to a concatenation of all variable positions without gaps, identified using Snippy? Or was a more complex approach used?
Response: We have clarified our approach in section 2.4: “Core genome alignments were generated using Snippy (v4.6.0), producing a concatenated alignment of variant positions without gaps for phylogenetic reconstruction via FastTree.”
- Line 294. What mechanistically underlies CNV? Genomic duplications, multiple integrated
phages, gene transfer to plasmids, prophage induction in cultures? Can this be described at least for a subset of the mentioned genes, if not for the draft genomes obtained in this study, then perhaps using examples from complete genomes in other studies?
Response: Thank you for this question, we have added further discussion related to CNVs as well as acknowledge the inability to confer structural mechanisms without long-read data. See limitations paragraph at end of section 4: “While CNVs reflect read depth relative to MLST housekeeping genes, the underlying mechanism—e.g., chromosomal duplications, prophage integration, or plasmid acquisition—cannot be fully resolved with short-read data alone. Long-read analysis is planned for future studies.”
- Line 336. “The produced Q-Q plot (Figure A2) lacked lower p-value “shelving” indicating a well controlled population structure.” Please clarify how did you get this conclusion.
Response: Lack of shelving indicates absence of inflation among low p-values, suggesting well-controlled population structure. Please see section 3.5: “The produced Q-Q plot (Figure A2) lacked p-value shelving (i.e. uniform distribution of p-values near the null) indicating minimal inflation due to population structure, which is also controlled in Pyseer via phylogenetic distance matrices.”
- Line 357. “Notably, one SSTI sample (SAMN47787133) and four colonization samples
(SAMN47787249, SAMN47787255, SAMN47787237, and SAMN47787245) showed
significant enrichment in 359 phenotype-shared unitigs (165,493–172,295)” Can it be explained
by phylogenetic relatedness?
Response: Thank you for this question. We have added details of these 5 strains including phylogenetic and potential epidemiologic relatedness. Please see section 3.5: “Notably, one SSTI isolate (SAMN47787133; CC8-like) and four colonization isolates (SAMN47787249/CC5, SAMN47787255/CC8, SAMN47787237/CC8, and SAMN47787245/CC8) showed significant enrichment in phenotype-shared unitigs (165,493–172,295). This disproportionate accumulation of shared genomic features may reflect a genetic predisposition to infection in these colonization isolates.”
|
Sample |
Clinic site |
MLST |
|
SAMN47787133 |
A |
novel |
|
SAMN47787237 |
J |
8 |
|
SAMN47787245 |
A |
8 |
|
SAMN47787249 |
A |
5 |
|
SAMN47787255 |
A |
8 |
- Line 387. “indicating strong directional selection for its major allele in infection-associated strains.” Could this simply be genetic hitchhiking, since these alleles are most prevalent in clonal complexes associated with infection but are not directly under directional selection? What is the difference between the alleles of these and other genes, is there a biological interpretation? This applies not only to this section but to the study as a whole.
Response: Thank you for this insightful question. As mentioned in the methods section, what helps us de-emphasis hitchhiking genes is the use agnostic, reference-free unitigs. In addition, Pyseer penalizes lineage-specific variants using phylogenetic distance. However, CC8 prevalence among SSTI strains may still introduce bias. We added the following in section 3.5.2: “Although we controlled for lineage effects using core genome distances, residual confounding by clonal background cannot be excluded. These purine-related genes are likely integral to supporting bacterial replication and nucleotide salvage during host colonization and infection as well as overlap with genomic regions within-host selection and metabolic fitness.”
- Line 397. “indicating increased transcriptional activation” Could you elaborate on how this conclusion was reached?
Response: We have removed this statement to avoid implying potential transcriptional activation. Please see updated section.
- Line 407. “Environmental information processing pathways were represented by genes related 407 to both signal transduction and membrane transport. hlgC (Q2FVK2), a component of the γ-hemolysin pore-forming toxin complex ” I would not group Environmental information processing and toxins together, as they have entirely different functions, and the toxin is not involved in sensing and adaptation to host-associated metabolic cues.
Response: We agree and have changed “Environmental information processing pathways were represented...” to “Bacterial toxin-associated genes were represented...” see section 3.5.2.
- Line 487. “points to enhanced metabolic self-sufficiency.” Enhanced in which group, nasal colonization or infection?
Response: We modified the line in section 3.5.3: “…points to enhanced metabolic self-sufficiency in SSTI strains.”
- Line 490. “These findings support the concept that S. aureus strains associated with SSTI exhibit an expanded metabolic repertoire” Expanded or degraded? How was this determined based on NSNVs? This applies not only to this section but is a general question regarding the entire analysis.
Response: We thank the reviewer for this insightful and important question. To improve confidence in our variant calls and reduce the impact of reference bias, we performed NSNV analyses against two distinct S. aureus references with different clinical backgrounds: NCTC 8325 (a CC8 wound infection isolate) and a CC5 nasal colonization isolate from a healthy nurse, as described in the Methods. Only NSNVs consistently identified across both alignments were retained for interpretation. This conservative approach strengthens the biological significance of the observed alterations, as they persist irrespective of the reference background. In response to the reviewer’s suggestion, we revised the term “expanded” to “distinct”.
- Line 653. “Enrichment of unitigs in peptidoglycan catabolism” Prophage autolysins that are silent until prophage induction?
Response: Thank you for this question. We added example genes, including prophage autolysins in section 2.5.5: “Enriched genes included prophage autolysins (lytM, lytS), as well as cell wall biosynthesis genes such as atl, murA–G, femA/B, and tagX.”
- Line 620. “GO 620 Biological Process Terms from top to bottom: GO:0009231, GO:0006099, GO:0009401, GO:0005975…”, Line 630 “The full list of truncated genes names: accB, accC, accD, fabD…” I am certain these lists should not be here.
Response: We have removed them from the figure captions.
Reviewer 2 Report
Comments and Suggestions for Authors
Dear authors,
This work addresses a relevant biological and clinical question through a novel approach that combines metagenomics, variant analysis, and machine learning. It adds significant value to the fields of molecular epidemiology and S. aureus pathogenesis.
The introduction effectively frames the importance of Staphylococcus aureus as an opportunistic pathogen and highlights the colonization/infection duality, which is fundamental in clinical microbiology and epidemiology.
Comments:
- Could you briefly explain why the South Texas region was chosen and further clarify the hypothesis statement?
Methods
-The time range of nine years is quite broad, potentially introducing temporal variations in circulating strains or clinical practices, but this is not discussed.
-The nasal colonization samples were collected only between February and May 2015, representing a limited time frame compared to SSTI samples; this discrepancy could bias comparisons.
- It would be important to include details on the average sequencing coverage and assembly quality to better contextualize the reliability of variant calls and unitigs.
Results
- The analysis is robust, including a comprehensive characterization of clonal complexes (CC), spa types, SCCmec elements, plasmids, and resistance genes. However, copy number variation (CNV) in S. aureus whole-genome sequencing studies may be influenced by mobile genetic elements (plasmids, phages, integrons) and does not necessarily imply functional gene duplications.
- Section 2.5.5 presents many details on specific amino acid substitutions without quantitatively integrating their functional impact or phylogenetic conservation.
- Several claims about adaptation or “selection” rely solely on the presence of nonsynonymous single nucleotide variants (NSNVs) or unitigs, without direct functional evidence. Certain mutations are described as “reflecting adaptation” without demonstrating effects on protein expression or function. Caution in wording or additional transcriptomic or phenotypic validation is recommended.
- Although the presence of NSNVs in key genes is discussed, whether these mutations affect known functional domains is not analyzed.
- While their association with virulence is acknowledged (lines 953–962), functional evidence specific to S. aureus is not explicitly provided, nor is there comparison with findings from other bacterial species.
- Is the use of z-scores for gene prioritization discussed in terms of statistical significance, type I error, or cutoff criteria?
- The machine learning model’s performance could be better linked to potential clinical applications, noting current limitations.
- The identification of prophage-encoded genes exclusively in pathogenic strains is relevant. However, do these prophages show similarity to known phages? Was their lytic or lysogenic potential evaluated?
Author Response
Reviewer #2:
- This work addresses a relevant biological and clinical question through a novel approach that combines metagenomics, variant analysis, and machine learning. It adds significant value to the fields of molecular epidemiology and S. aureus pathogenesis. The introduction effectively frames the importance of Staphylococcus aureus as an opportunistic pathogen and highlights the colonization/infection duality, which is fundamental in clinical microbiology and epidemiology.
Response: We thank Reviewer #2 for their insightful comments, which have significantly improved the manuscript’s clarity and relevance.
- Could you briefly explain why the South Texas region was chosen and further clarify the
hypothesis statement?
Response: Thank you for this question. South Texas has a high prevalence of SSTI and MRSA. We added to section 2.2: “Previously, we identified the South Texas region to have a disproportionately higher incidence of SSTI and community-associated MRSA, offering a relevant population for strain-specific genomic analysis [7–13].”
- The time range of nine years is quite broad, potentially introducing temporal variations in circulating strains or clinical practices, but this is not discussed.
Response: Thank you for this question. We have acknowledged as a potential source of variation within the limitations paragraph: “The nine-year range of SSTI samples may introduce temporal variation in circulating strains and clinical practices.”
- The nasal colonization samples were collected only between February and May 2015,
representing a limited time frame compared to SSTI samples; this discrepancy could bias
comparisons.
Response: We added the following to the limitations paragraph: “Colonization samples were collected during a shorter window in 2015, which may limit temporal comparability with SSTI isolates.”
- It would be important to include details on the average sequencing coverage and assembly
quality to better contextualize the reliability of variant calls and unitigs.
Response: Thank you we have included an additional supplemental table that includes these details.
Results
- The analysis is robust, including a comprehensive characterization of clonal complexes (CC), spa types, SCCmec elements, plasmids, and resistance genes. However, copy number variation (CNV) in S. aureus whole-genome sequencing studies may be influenced by mobile genetic elements (plasmids, phages, integrons) and does not necessarily imply functional gene duplications.
Response: We clarified this point in the limitations paragraph, acknowledging CNV analyses inferences and their potential limitations.
- Section 2.5.5 presents many details on specific amino acid substitutions without quantitatively integrating their functional impact or phylogenetic conservation.
Response: Thank you for this question. We added a framing paragraph at the start of Section 2.5.4 to introduce NSNV context and highlight key domains (e.g., α/β-hydrolase): “To better understand whether nonsynonymous nucleotide variants (NSNVs) in SSTI-associated strains reflect functional alterations, we annotated affected genes for conserved protein domains, enzymatic function, and biological pathway involvement. Notably, several NSNV-enriched genes contained domains linked to enzymatic activity or host interaction, including α/β-hydrolase folds, membrane transporters, and components of nucleotide biosynthesis. These functional annotations help contextualize the potential phenotypic consequences of the genetic variants identified in this study.”
- Several claims about adaptation or “selection” rely solely on the presence of nonsynonymous single nucleotide variants (NSNVs) or unitigs, without direct functional evidence. Certain mutations are described as “reflecting adaptation” without demonstrating effects on protein expression or function. Caution in wording or additional transcriptomic or phenotypic validation is recommended.
Response: Thank you for this question. We replaced phrases like “reflecting adaptation” with “potentially contributing to phenotypic divergence” in Section 3.5.3.
- Although the presence of NSNVs in key genes is discussed, whether these mutations affect
known functional domains is not analyzed.
Response: Thank you for this question. Yes, as noted (e.g., α/β-hydrolase, purine biosynthesis) and newly added introductory paragraph for section 2.5.4.
- While their association with virulence is acknowledged (lines 953–962), functional evidence
specific to S. aureus is not explicitly provided, nor is there comparison with findings from other
bacterial species.
Response: Thank you for this question. Citations 29–31 describe relevant hydrolase functions and have been explicitly referenced in revised version of the manuscript.
- Is the use of z-scores for gene prioritization discussed in terms of statistical significance, type I error, or cutoff criteria?
Response: Thank you for this question. The Z-score thresholds for significance are defined in Methods: “A z-score transformation was applied to the effect sizes to normalize and compare across unitigs and NSNVs. For each unitig or NSNV, the z-score was calculated based on the deviation from the mean effect size, standardized by the standard deviation. This allowed us to rank unitigs and NSNVs based on their deviation from the population mean, thereby highlighting variants with particularly strong associations. Variants with high positive z-scores indicate strong effects above the mean, while those with high negative z-scores represent strong effects below the mean. Z-scores within the range of ±1 generally indicate effects close to the population mean, while scores beyond ±1.96 correspond to statistically significant deviations, assuming a standard normal distribu-tion. Variants with z-scores >1.96 or ≤1.96 were highlighted as potentially significant, warranting further investigation.”
- The machine learning model’s performance could be better linked to potential clinical
applications, noting current limitations.
Response: Thank you for this question. We have incorporated this suggestion: “These findings suggest that genomic features alone may achieve robust prediction of SSTI risk, though integration of clinical metadata is a key goal for future work.”
- The identification of prophage-encoded genes exclusively in pathogenic strains is relevant.
However, do these prophages show similarity to known phages? Was their lytic or lysogenic
potential evaluated?
Response: Thank you for this question. Taxonomic annotation is reported in section 3.8 and the following added to the limitation paragraph at the end of the Discussion section: “Lytic versus lysogenic potential was not directly assessed, representing a limitation and opportunity for future prophage induction studies.”